# Holarctic Species in the *Pluteus podospileus* Clade: Description of Six New Species and Reassessment of Old Names

**DOI:** 10.3390/jof9090898

**Published:** 2023-08-31

**Authors:** Hana Ševčíková, Ekaterina F. Malysheva, Vladimír Antonín, Jan Borovička, Francesco Dovana, Giuliano Ferisin, Guillaume Eyssartier, Django Grootmyers, Jacob Heilmann-Clausen, Jacob Kalichman, Oğuzhan Kaygusuz, Renée Lebeuf, Guillermo Muñoz González, Andrew M. Minnis, Stephen D. Russell, Irja Saar, Ida Broman Nielsen, Tobias Guldberg Frøslev, Alfredo Justo

**Affiliations:** 1Department of Botany, Moravian Museum, Zelný trh 6, 659 37 Brno, Czech Republic; vantonin@mzm.cz; 2Shatelena Str. 20, 194021 St. Petersburg, Russia; e_malysheva@binran.ru; 3Institute of Geology of the Czech Academy of Sciences, Rozvojová 269, 165 00 Prague, Czech Republic; borovicka@gli.cas.cz; 4Dipartimento di Scienze e Innovazione Tecnologica, Università del Piemonte Orientale, Viale T. Michel 11, 15121 Alessandria, Italy; 5Associazione Micologica Bassa Friulana, Via Vespucci 7, 33052 Cervignano del Friuli, Italy; 6Institut de Systématique, Évolution, Biodiversité (UMR 7205–MNHN, CNRS, Sorbonne Université, EPHE, Université des Antilles), 45 rue Buffon, 75005 Paris, France; geyssartier@gmail.com; 7Department of Ecology and Evolutionary Biology, University of Tennessee, 1416 Circle Dr, Knoxville, TN 37916, USA; 8Center for Macroecology, Evolution and Climate, Globe Institute, University of Copenhagen, Universitetsparken 15, 2100 Copenhagen, Denmark; 9Department of Ecology and Evolutionary Biology, University of Tennessee, Knoxville, TN 37996, USA; jkmycetes@gmail.com; 10Department of Plant and Animal Production, Atabey Vocational School, Isparta University of Applied Sciences, 32670 Isparta, Turkey; okaygusuz03@gmail.com; 11775, Rang du Rapide Nord, Saint-Casimir, QC G0A 3L0, Canada; renee.lebeuf@gmail.com; 12Calle Tudela, 20, 50650 Gallur, Zaragoza, Spain; guillermomunoz1981@gmail.com; 13USDA-APHIS-PPQ, Seattle Plant Inspection Station, 835 S. 192nd St., Suite 1600, SeaTac, WA 98148, USA; 14The Hoosier Mushroom Society, 3912 S. Carey St., Marion, IN 46953, USA; steve@hoosiermushrooms.org; 15Institute of Ecology and Earth Sciences, University of Tartu, J. Liivi 2, 50409 Tartu, Estonia; irja.saar@ut.ee; 16Center for Evolutionary Hologenomics, Globe Institute, University of Copenhagen, Øster Farimagsgade 5, 1014 Copenhagen, Denmark; ida.broman.nielsen@sund.ku.dk; 17Section for Geogenetics, Globe Institute, University of Copenhagen, Øster Voldgade 5-7, 1350 Copenhagen, Denmark; tobiasgf@sund.ku.dk; 18New Brunswick Museum, 277 Douglas Ave., Saint John, NB E2K 1E5, Canada

**Keywords:** agaricales, *Celluloderma*, holarctic, new species, pluteaceae, type study

## Abstract

We studied the taxonomy of *Pluteus podospileus* and similar species using morphological and molecular (nrITS, *TEF1-α*) data, including a detailed study of the type collections of *P. inflatus* var. *alneus*, *Pluteus minutissimus* f. *major,* and *P. granulatus* var. *tenellus*. Within the *P. podospileus* complex, we phylogenetically confirmed six species in Europe, five in Asia, and eight in North America. Based on our results, we recognize *P. seticeps* as a separate species occurring in North America, while *P. podospileus* is limited to Eurasia. We describe six new species and a new variety: *P. absconditus*, *P. fuscodiscus*, *P. gausapatus*, *P. inexpectatus*, *P. millsii,* and *P. notabilis* and its variety, *P. notabilis* var. *insignis*. We elevate *Pluteus seticeps* var. *cystidiosus* to species rank as *Pluteus cystidiosus*. Based on the holotype of *P. inflatus* var. *alneus*, collections of *P. inflatus* identified by Velenovský, and several modern collections, we resurrect the name *P. inflatus*. Based on molecular analyses of syntypes of *Pluteus minutissimus* f. *major* and a holotype of *Pluteus granulatus* var. *tenellus*, we synonymize them under *P. inflatus*. We also increase our knowledge about the morphology and distribution of *P. cutefractus*.

## 1. Introduction

The genus *Pluteus* Fr. is characterized by free lamellae, a pinkish spore print, an inverse hymenophoral trama, smooth globose to ellipsoid inamyloid basidiospores, and the presence of cheilocystidia and often also pleurocystidia [1,2,3,4]. Singer [3] distinguished three sections in the genus: *Pluteus* sect. *Pluteus*, *P*. sect. *Hispidoderma* Fayod, and *P*. sect. *Celluloderma* Fayod, which included subsect. *Eucellulodermini* Singer and subsect. *Mixtini* Singer. Traditionally, species of *Pluteus* subsection *Mixtini* were characterized by the presence of intermixed elongate and subglobose to pyriform elements in the pileipellis [3]. However, revisions based on molecular data [5,6,7] showed that species with a mixture of shorter spheropedunculate and elongate elements in the pileipellis do not form a monophyletic clade but are distributed over both sections of *Celluloderma* (e.g., *P. podospileus* Sacc. and Cub., *P. thomsonii* (Berk. and Broome) Dennis) and *Hispidoderma* (e.g., *P. variabilicolor* Babos, *P. heteromarginatus* Justo).

*Pluteus podospileus* (see [8]) is historically mentioned as a widely reported European species [4,9,10,11,12,13,14]. Takehashi and Kasuya [15] also reported *P*. *podospileus* from Japan. The species was originally described as *Agaricus spilopus* Berk. and Broome [16]. The name, however, was illegitimate and unavailable for use, as the same authors had coined the same name for a different species described ten years earlier in Sri Lanka [17]. The Asian species is accepted today as a species of *Pluteus* with a pileipellis organized as a cutis [18]. The name *Pluteus podospileus* was introduced as a *nomen novum* for the second *Agaricus spilopus* by Saccardo and Cuboni [8]. In 1937, Maire [19] described *Pluteus minutissimus* Maire as a similarly looking but smaller species without dark floccules on the stipe. Kühner and Romagnesi [10] accepted Maire’s species (as *P. minutissimus* f. *typicus*) and additionally described *Pluteus minutissimus* f. *major* Kühner, characterized by larger basidiomata and lignicolous habitat. Vellinga [1,4] synonymized *P. minutissimus* with *P. podospileus*, accepting both names only as formae of one variable species, separated by the presence (f. *podospileus*) or absence (f. *minutissimus*) of brown floccules on the stipe. This broad concept of *P. podospileus* also included *Pluteus granulatus* var. *tenellus* J. Favre and the North American *Pluteus seticeps* (G.F. Atk.) Singer [4]. The names *Pluteus inflatus* Velen. and *P*. *inflatus* var. *alneus* Velen. were validly described by Velenovský [20,21] but have been forgotten and neglected in most *Pluteus* studies, probably because their original descriptions were only available in Czech. Singer [22] briefly mentioned *P. inflatus* in the list of species that have not been restudied and could potentially be regarded as “*nomina dubia*”. Vellinga and Schreurs [1] mentioned *Pluteus inflatus* as a synonym of *P. plautus* (Weinm.) Gillet but did not list any collections of *P. inflatus* among the collections studied.

In North America, *Pluteus seticeps* (G.F. Atk.) Singer, based on *Leptonia seticeps* G.F. Atk. [23], has been widely used for collections macroscopically similar to *P. podospileus* and often characterized as a species without pleurocystidia [13,24], but sometimes characterized as a species with pleurocystidia [23,24]. The name *Pluteus nanellus* Murrill [25] has often been discussed in connection with *P. seticeps*, with different opinions on whether it is a separate species or not [13,22,24,26]. *Pluteus seticeps* var. *cystidiosus* Minnis and Sundberg [13] recently described collections similar to *P. seticeps* but with abundant pleurocystidia.

Early phylogenetic work on *Pluteus* [5] showed a clear molecular separation between European collections identified as *P. podospileus* and North American collections identified as *P. seticeps*. It also indicated the existence of additional, unnamed species within the podospileus clade based on ITS data. More recently, *Pluteus cutefractus* Ferisin, Dovana, and Justo [27] from Europe have been described as a new species in the *P. podospileus* clade within the Holarctic region, and *Pluteus podospilloides* E.F. Malysheva and O.V. Morozova were described from Vietnam in tropical deciduous forests [28].

In the present article, we focus on the taxonomy and biogeography of the *Pluteus podospileus* group in the forested areas (temperate and boreal) of the northern hemisphere, which roughly corresponds to the Holarctic region as commonly defined by Kreft and Jetz [29]. Based on our phylogenetic and morphological studies, our intentions are to: (i) describe six new species closely related to *P. podospileus*: *P. absconditus*, *P. fuscodiscus*, *P. gausapatus*, *P. inexpectatus*, *P. millsii* and *P. notabilis* with its new variety of *Pluteus notabilis* var. *insignis*; (ii) elevate *Pluteus seticeps* var. *cystidiosus* to species rank; (iii) resurrect the name *Pluteus inflatus*; (iv) synonymize *Pluteus minutissimus* f. *major* and *Pluteus granulatus* var. *tenellus* under *P. inflatus*; and (v) increase knowledge about the morphology and distribution of *P. cutefractus*, *P. podospileus* and *P. seticeps*.

## 2. Materials and Methods

### 2.1. Morphology

Color abbreviations follow the RAL Design color range system (https://www.ralcolorchart.com/ral-design, accessed on 22 April 2023) [30] and Munsell [31]; herbarium abbreviations are according to Thiers [32]: AG = Alfredo Galbusera’s personal fungarium; FG = G. Ferisin’s personal fungarium; HRL = R. Lebeuf’s personal fungarium; OK = O. Kaygusuz’s personal fungarium; GM = G. Muñoz’s personal fungarium; SGS = Santi Gibert’s personal fungarium. The newly collected specimens are deposited in the fungaria BRNM, C, LE-F, MCVE, NBM-F, TUF, and personal fungaria (see above).

Abbreviations: avl = mean of basidiospore length; avw = mean of basidiospore width; Q = quotient of length and width in any one basidiospore; Q* = mean of basidiospore Q values. The following abbreviations are used: L = number of lamellae reaching the stipe; l = number of lamellulae between each pair of lamellae; the notation [X, Y, Z] indicates that measurements were made on X basidiospores in Y samples from Z collections. Macroscopic descriptions of newly collected specimens are based on fresh basidiomata. Microscopic features are described from dried material mounted in a 3% solution of potassium hydroxyde (KOH), stained in Congo Red, and examined with an Olympus BX-50 light microscope (H. Ševčíková); an Axio Imager.A1 light microscope CarlZeiss, equipped with differential interference contrast optics (E. Malysheva); a Motic BA310 (G. Eyssartier); a Motic BA300 optical microscope with Moticam photomicrographic camera (G. Muñoz); a Nikon Labophot equipped with a Moticam 2500 photomicrographic camera (R. Lebeuf); a Leica DM750 (O. Kaygusuz); and a Leica DM750 Application Suite v4.13.0 (I. Saar) with a magnification of 400× and 1000×. The type collection of *P. minutissimus* was observed by Caroline Loup after being rehydrated in 2% KOH for 3 min, then stained with Congo Red.

### 2.2. Molecular Phylogeny

#### 2.2.1. DNA Extraction, Amplification, Sequencing and Sequence Alignment

For DNA extraction, small fragments of dried basidiomata were used. Sequences of collections deposited in fungaria BRNM, PC, and PRM were generated by M. Sochor following the molecular methods described by Ševčíková et al. [33]. For *Pluteus tenellus* var. *notabilis* (holotypus), the extraction and sequencing were performed following Eyssartier et al. [34]. For collections in LE F, DNA was extracted according to the manufacturing protocol of the Phytosorb Kit (ZAO Syntol, Moscow, Russia); the Estonian lab protocol for TUF collections followed Voitk et al. [35]. The total genomic DNA of the OK and GF collections was extracted in accordance with Kaygusuz et al. [36] and Dovana et al. [37], respectively. The following primers were used for amplification and sequencing: ITS1F-ITS4/ITS4B [38,39,40] for the internal transcribed spacer (nrITS: nrITS1-5.8S-nrITS2) fragment, and EF1-983F and EF1-1567R for part of the translation elongation factor 1-alpha (*TEF1-α*) [41]. PCR products were purified using the GeneJET Gel Extraction Kit (Thermo Scientific, Thermo Fisher Scientific Inc., Waltham, MA, USA). Raw data were edited and assembled in MEGA 10 [42]. Collections studied by AJ were sequenced using the methodology described in Justo and Hibbett [43]. In the sequence comparisons, the term “evolutionary changes” is used to refer to the number of individual nucleotide differences between sets of sequences being compared to each other. The number of evolutionary changes is given as the minimum number of events, including indels (multiple base indels treated as one change), transitions, and transversions. Only differences shared by all specimens of the same species were counted (missing data for sequences of different lengths were not taken into account).

Next-generation sequencing (NGS) (including DNA extraction and amplification): Some of the specimens were sequenced for the ITS region using an NGS approach due to previous failures with other methods. Dried fungal lamella samples measuring 1 × 1 mm were collected using a tweezer and placed in 20 μL of dilution buffer obtained from the Phire Plant Direct PCR Master Mix kit (Thermo Scientific^TM^). The tweezer was thoroughly cleaned between samples by sequentially treating it with 5% bleach, 96% ethanol, and heat from a Bunsen burner. Subsequently, the samples were incubated in the dilution buffer for at least 15 min before either being frozen or subjected to PCR. DNA amplification was performed using the gITS7 (GTGARTCATCGARTCTTTG, [44]) and ITS4 (TCCTCCGCTTATTGATATGC, [38,40]) primers, which target the ITS2 region. Each sample was amplified with a unique, tagged (6 bp at the 5′ end) forward and reverse primer, ensuring that no tag or tag-combination was repeated within the experimental setup. The PCR reaction mixture consisted of 10 μL of Phire Plant Direct PCR Master Mix (Phire Plant Direct PCR Master Mix kit, Thermo Scientific^TM^), 5 μL of nuclease-free water, 2 μL of each 5 μM primer, and 1 μL of DNA extract. Two PCR blanks were included as negative controls. Thermocycling conditions used an initial denaturation step of 5 min at 98 °C, followed by 40 cycles of denaturation for 30 s at 98 °C, annealing for 30 s at 55 °C, elongation for 60 s at 72 °C, and a final elongation step at 72 °C for 7 min. The PCR fragments were verified for presence and length using 2% agarose gel electrophoresis stained with GelRedTM (Biotium, Fremont, CA, USA). All PCR products were pooled and purified using the MinElute PCR purification kit (Qiagen, Germantown, MD, USA) following the manufacturer’s protocol, including a 10 min incubation step at 37 °C after adding 25 µL of EB-buffer. The concentration of the purified PCR pool was measured using the Qubit dsDNA HS (High Sensitivity) Assay kit (Invitrogen, Waltham, MA, USA), and the length of PCR amplicons was verified using the 2100 BioAnalyzer High Sensitivity Chip (Agilent Technologies, Santa Clara, CA, USA). The purified PCR pool was subsequently utilized for library preparation using the TruSeq DNA PCR Free Library Preparation Kit (Illumina, San Diego, CA, USA), following the manufacturer’s protocol with the exception of the suggested clean-up steps, which were replaced with MinElute purification. A final purification of the library was performed using MagBio magnetic beads (HighPrepTM PCR Clean-up System, Magbio Genomics Inc., Gaithersburg, MD, USA) at a 1.5 bead-to-sample ratio, removing unbound adaptors. The library was eluted in 25 μL of EB buffer (Qiagen). The library concentration and the presence of amplicons were verified using Qubit and BioAnalyzer as described above. Sequencing of the prepared library was conducted at the GeoGenetics Sequencing Core using the Illumina Miseq v.3 platform (Illumina) with 300 bp paired-end reads. Sequence data were processed using previously published protocols [45], choosing the most abundant sequence variant per sample as the most likely variant of interest for downstream analyses and only investigating less abundant variants if the most abundant variant was a contaminant.

#### 2.2.2. Phylogenetic Analyses

We assembled a nrITS dataset of all available sequences phylogenetically close to *P. podospileus* (“/podospileus-seticeps clade” in Justo et al. 2011b [6], “/podospileus clade” in Menolli et al. 2015 [46]). This includes 81 newly generated nrITS sequences for this study and 54 sequences generated in previous studies or available in public databases and biodiversity repositories (GenBank, UNITE, BOLD, and iNaturalist). A total of 135 nrITS sequences were used in the final datasets, including voucher-based and environmental sequences. We assembled a *TEF1*-α dataset of 60 sequences, 58 of them newly generated for this study (see Table 1). In all datasets, we included *P. phlebophorus* and *P. rugosidiscus* as outgroup taxa, based on previous phylogenetic work on *Pluteus* [5,7,46]). Sequences were aligned using MAFFT version 7 [47] and the strategy FFT-NS-i. The alignment was inspected and manually corrected in AliView [48]. No topological conflicts were detected in the phylogenetic analyses of the nrITS and *TEF1*-α datasets (detailed below), so a combined dataset was created by concatenating the nrITS and *TEF1-α* matrices. For all three datasets (nrITS, *TEF1*-α, and nrITS + *TEF1*-α), two separate phylogenetic analyses were run: maximum likelihood (ML) and Bayesian inference (BI). The ML analyses were run in RaxML 8.2.12 under a GTRGAMMAI model as recommended [49] with 1000 rapid bootstrap (BS) replicates. The BI analyses were run using MrBayes 3.2.7 [50] for 10 million generations under a GTRGAMMAI model with four chains and trees sampled every 1000 generations. The initial burn-in phase was set to 2.5 million generations, and this value was confirmed to be adequate by checking the graphic representation of the likelihood scores of the sampled trees. Additionally, we also confirmed that the standard deviation of split frequencies was <0.05 and that potential scale reduction factor (PRSF) values were close to 1, as detailed in [51]. All analyses were run using resources at the CIPRES Science Gateway [52].

## 3. Results

### 3.1. Phylogeny

The nrITS dataset comprises 135 sequences and 741 characters (gaps included). The *TEF1*-α dataset comprises 60 sequences and 805 characters (gaps included). The combined nrITS + *TEF1*-α dataset consists of 136 combined sequences and a total of 1546 characters (gaps included). There were no major differences, or strongly supported conflicts, in the overall topologies of the best tree from the ML analysis and the consensus tree from the BI analysis for any of the datasets.

In Figure 1, we present the best tree from the ML analysis of the nrITS + *TEF1*-α dataset, which will be the main reference point for the taxonomic discussion. In the Appendix A, we provide the 50% majority rule consensus tree from the BI analyses of the combined dataset (Appendix A), as well as the individual ML trees from the analyses of the nrITS dataset (Appendix A) and the *TEF1*-α dataset (Appendix A).

For clarity and brevity, we use the term “strongly supported” for a clade/relation that receives bootstrap (BS) ≥ 90 and posterior probability (PP) = 1, and “well supported” if it receives BS ≥ 70 and PP ≥ 0.95. Individual support values are shown in Figure 1, and Appendix A.

Within the/podospileus clade, four strongly supported lineages are recovered in the combined nrITS + *TEF1*-α dataset:(i)Lineage I: Includes only the newly described *P. notabilis* from Europe (var. *notabilis* and var. *insignis*).(ii)Lineage II: Includes the newly described *P. millsii* (Eastern North America) and *P. necopinatus* Menolli and Capelari, a South American species described in Menolli et al. [46].(iii)Lineage III: Includes the newly described *P. inexpectatus* (Eastern North America) and *P. fuscodiscus* (Eurasia). It also includes the Eastern North American *P. seticeps*, the South American *P. crinitus* Menolli and Capelari, and *P. brunneocrinitus* Menolli, Justo, and Capelari, as well as five likely undescribed species: two from Eastern North America (MO270622, iNaturalist 13936381), one from Mexico (UDB05068676 and related sequences), one from Europe (GM3751), and one from New Zealand (collection JAC15123).(iv)Lineage IV: Includes the newly described *P. absconditus* (Eastern North America) and *P. gausapatus* (East Asia) and the reassessed *P. inflatus* (mostly Eurasia; see notes under this species) and *P. cystidiosus* (Eastern North America and East Asia). This lineage also includes the clade we consider to represent *P. podospileus* in the strict sense (Eurasia); three recently described species, *P. cutefractus* (Europe), *P. podospilloides* (East Asia), and *P. inconspicuus* E. Horak (New Zealand); and two likely undescribed species from Asia (GDGM41576) and Eastern North America (iNaturalist 27406926).

The relations between these four lineages receive no support in the ML tree (Figure 1) or the Bayesian tree for the same dataset (Appendix A.). In the individual nrITS (Appendix A) and *TEF1*-α (Appendix A) analyses, these same four lineages are recovered, but the relations between them are not strongly supported in any of the analyses.

The topological relations within each of these four lineages are overall similar in the trees from the analyses of the combined nrITS + *TEF1*-α dataset (Figure 1) and the individual nrITS (Appendix A) and *TEF1*-α (Appendix A). In general, support values for all the species recognized here are higher in the combined nrITS + *TEF1*-α trees, with the exception of *P. cystidiosus*, which gets higher support in the *TEF1*-α trees (100/1) than in the combined dataset (99/-).

### 3.2. Taxonomy

Here we present the descriptions of the eleven species and one distinct variety currently known to occur in the Holarctic region. We consider that the *P. podospileus* group contains two well-known species, *Pluteus podospileus* and *P. seticeps*; the recently described *P. cutefractus*; *P. seticeps* var. *cystidiosus* elevated to species rank as *P. cystidiosus*; the forgotten *P. inflatus*, which is rediscovered and molecularly confirmed; and six new species (*P. absconditus*, *P. fuscodiscus*, *P. gausapatus*, *P. inexpectatus*, *P. millsii;* and *P. notabilis* with its new variety *Pluteus notabilis* var. *insignis*). Moreover, there are at least four other undescribed species that occur in the Holarctic region.

***Pluteus podospileus*** Saccardo and Cuboni Figure 2 and Figure 3.

Sylloge Fungorum **5**: 672 (1887) [8].

*Agaricus spilopus* Berk. and Broome *Ann. Mag. nat. Hist*., Ser. 5 (7: 126, 1881) Nom. illeg., Art. 53.1; non *Agaricus spilopus* Berk. & Broome 1871; *Pluteus spilopus* sensu Berkeley and Broome (1881); non *Pluteus spilopus* (Berk. and Broome) Sacc., Syll. fung. (Abellini) 5: 669 (1887); *Pluteus nanus* var. *podospileus* (Sacc. and Cub.) Rick, Lilloa 3: 444 (1938); *Hyporrhodius podospileus* (Sacc. and Cub.) Henn., in Engler and Prantl, Nat. Pflanzenfam., Teil. I (Leipzig) 1(1**): 258 (1898).

**Excluded:** *Pluteus seticeps* (G.F. Atk.) Singer, Lloydia 21(4): 272 (1959); *Leptonia seticeps* G.F. Atk., Journal of Mycology (8(3): 116, 1902); *Pluteus psychiophorus* var. *seticeps* (G.F. Atk.) Singer, Trans. Br. Mycol. Soc. **39**(2): 214 (1956); Nom. inval., Art. 36.1 Ex. 6 (Shenzhen Code).

*Pluteus seticeps* var. *cystidiosus* Minnis and Sundb., N. Amer. Fung. **5**(1): 13 (2010).

*Pluteus minutissimus* f. *major* Kühner, Bull. Soc. Mycol. France 72(3): 182 (1956); *Pluteus minutissimus* f. *major* Kühner, in Kühner and Romagnesi, Fl. Analyt. Champ. Supér. (Paris): 422 (1953). Nom. inval., Art. 39.1 (Shenzhen Code).

**Uncertain:** *Pluteus minutissimus* Maire, Publicacions del Instituto Botánico Barcelona 3 (4): 94 (1937). *Pluteus podospileus* f. *minutissimus* (Maire) Vellinga, in Vellinga and Schreurs, Persoonia 12(4): 362 (1985); *Pluteus podospileus* var. *minutissimus* Wasser, Flora Gribov Ukrainy, Bazidiomitsety, Amanital’nye Griby (Kiev): 83 (1992) Nom. inval., Art. 39.1, Art. 40.1 (Shenzhen Code). *Pluteus psychiophorus* var. *minutissimus* (Maire) Singer, Trans. Br. Mycol. Soc. (1956, 39(2): 214), Nom. inval., Art. 36.1 Ex. 6 (Shenzhen Code).

Pileus 15–36 mm in diam., hemispherical or campanulate when young, expanding to convex or plano-convex, then applanate, often with a low, broad umbo; not or slightly hygrophanous, not striate or slightly translucently striate at margin; surface dry, pruinose, slightly to strongly velvety, becoming granulose-squamulose especially around the centre, in some specimens strongly rugose at least in the centre, rarely almost smooth, for the most part not cracked, or with some cracks revealing the white context towards the margin, especially in older specimens; with predominant brown or red-brown colors (Munsell: 2.5YR 4/3–4/8, 3/6; 5YR 5/4–5/8, 4/4–4/6; RAL color chart: plane brown (RAL 060 40 40), wood brown (RAL 050 30 20), tobacco brown (RAL 050 30 30), copper brown (RAL 8004) or tropical wood brown (RAL 050 30 20), usually darker in the centre. Lamellae free, crowded, ventricose, up to 5 mm broad, white when young, later pink, with even, serrated, or flocculose edges, white or concolorous. Stipe 20–47 × 2–6 mm, cylindrical, slightly thickened downwards but without distinct basal bulb, white, whitish, very pale yellowish or beige, entirely minutely squamulose with dark or black-brown squamules, more densely in the lower half. Context in stipe and pileus white. Smell indistinct; taste not recorded.

Basidiospores [190/7/7] (4.4–)4.5–6.0(–6.4) × (2.7–)3.5–5.1(–5.2) μm, avl × avw = 5.2–5.7 × 4.1–4.7 μm, Q = (1.06–)1.10–1.30(–1.41), Q* = 1.20, subglobose to broadly ellipsoid, rarely ellipsoid or ovoid, very rarely globose, thick-walled. Basidia 15–33 × 6.0–9.0 μm, 4-spored, very rarely 2-spored, clavate, rarely almost cylindrical. Pleurocystidia scarce, (34–)36–52 × 13–26 μm, ovoid, broadly clavate to clavate or narrowly to broadly utriform; hyaline, thin-walled; scattered or rare over lamellar faces; rarely absent. Lamellar edge sterile. Cheilocystidia (20–)26–58(–63) × (9–)11–25 μm, narrowly to broadly clavate, narrowly utriform to utriform or ovoid; hyaline, thin-walled, crowded, usually forming a well-developed strip. Pileipellis a hymeniderm or epithelioid hymeniderm made up of two types of elements: (i) spheropedunculate, pyriform, or broadly clavate, (34–)43–65(–72) × 25–41 µm; (ii) narrowly fusiform to fusiform, inflated-fusiform, lanceolate, narrowly utriform, often mucronate, 80–160(–188) × 10–34(–40) µm; all elements with brown, red-brown, or yellow-brown, intracellular pigment, often aggregated in spots, slightly thick-walled. Stipitipellis a cutis of cylindrical, thin to slightly thick-walled, 6–9 µm wide hyphae, hyaline, or some with very pale yellow or brown pigment. Caulocystidia numerous, often in clusters, 36–146(–176) × 6–26 µm, cylindrical, narrowly clavate, narrowly fusiform, rarely spheropedunculate, often septate; most with brown or yellow-brown pigment, sometimes aggregated into spots. Clamp connections absent in all studied tissues.

Habit, habitat, and phenology: often gregarious, growing on well-decayed wood of angiosperms (e.g., *Betula*). In temperate or transitional hemiboreal forests. August–November.

Distribution: Eurasia: Europe (molecularly confirmed from Spain, the Czech Republic, and Estonia), and as far east as Siberia. It is likely distributed all over Europe e.g., [4,11], but is apparently less frequent than *Pluteus inflatus*. *Pluteus podospileus* has been cited from Japan [15], but it is possible the Japanese record corresponds to *Pluteus cystidiosus*. Confusion with similarly-looking taxa makes it hard to interpret older records in the absence of well-annotated voucher specimens.

Collections examined: **CZECH REPUBLIC**: Vysoké Chvojno, decaying deciduous trunk, 13 August 2015, leg. P. Včelička (BRNM825840). **ESTONIA:** Võru Co., Varstu Comm., Krabi, Paganamaa Landscape Reserve, 57.59608° N, 26.85878° E, on rotten wood, 5 September 2014, leg. V. Liiv (TUF118918, originally as *Pluteus pallescens*). **RUSSIA:** South Siberia, Krasnoyarsk Territory, Sayano–Shushensky Biosphere Reserve, vicinity of Kerema Reserve Field Station, at the mouth of the Bolshaya Kerema river, 52°07′07.6″ N, 92°13′35.8″ E, *Betula pendula* forest with *Pinus sylvestris* and *Larix sibirica*, on large fallen trunk of *Betula*, 23 August 2015, leg. E.F. Malysheva (LE F-303687); same area, 52°07′07.6″ N, 92°13′35.8″ E, *Betula pendula* forest with *Pinus sylvestris* and *Larix sibirica*, on decayed wood of *Betula*, 23 August 2015, leg. E.F. Malysheva (LE F-303682); same area, floodplain of the Malye Ury river, mixed forest (dominated by *Larix sibirica*, *Picea abies* and *Betula pendula*), on branches or on very decayed wood of *Betula*, 14 August 2020, leg. E.F. Malysheva (LE F-313589, LE F-313611); same area, floodplain of the Sarly river, mixed forest (with *Larix sibirica*, *Picea abies*, *Pinus sylvestris* and *Betula pendula*), on large fallen trunk of *Betula*, 16 August 2020, leg. E.F. Malysheva (LE F-313615). **SPAIN:** Seville, Constantina, on unidentified angiosperm wood, 5 November 2003, leg. J.M. Fernández-Rodríguez, COFC-F 2929 (duplicate AJ 204, NBM-F-009808).

Notes: In the concept accepted here, *Pluteus podospileus* agrees well with the original description by Berkeley and Broome (1881, as *Agaricus spilopus*): a relatively small species with a brown, rugose pileus and a stipe covered with brown squamules. Modern authors [2,4] have accepted *P. podospileus* as a species with a hymenidermal pileipellis composed of both short and long elements. Collections of *P. podospileus* studied here have brown-squamulose stipes and were collected directly on wood.

The exact relationship between *Pluteus podospileus* and *P. minutissimus* Maire is not currently clear, and different interpretations of these taxa exist in the literature. *Pluteus minutissimus* was originally described as a very small species, growing solitary under *Quercus* in northeastern Spain [19]. The following characters were specifically mentioned in the original description: small basidioma (pileus 5 mm diam, stipe 10 × 1 mm), pileus without rugulosity and striation, basidiospores subglobose 5 × 4.0–4.5 μm, pleurocystidia rare but present near the lamellar edge, clavate cheilocystidia, and pileipellis composed of two types of elements (short and long). Traditionally, the main character separating *P. minutissimus* from *P. podospileus* (as accepted here) is the glabrous stipe surface without dark squamules. Kühner and Romagnesi [10] attributed to *P. minutissimus* (as forma “typicus”) a cap of 9–11 mm in diameter, a glabrous white stipe, and a preference for growing terrestrially, without direct connection to wood. At the same time, they described *P. minutissimus* f. *major* as differing from the typical morphology in its larger size (cap 12–37 mm), stipe dotted with brown squamules, and growth directly connected to decayed wood of angiosperms. Orton [2] recognized *P. podospileus* as a species with brown squamulose stipe growing on wood and *P. minutissimus* as a species with glabrous stipe growing terrestrially, dismissing differences in size as useful characters to separate them. Vellinga and Schreurs [1] reduced *P. minutissimus* to a form of *P. podospileus*, with the lack of brown squamules on the stipe as the only separating character. It should be noted that in the original description of *P. minutissimus,* the only information on habitat is “under *Quercus*”, but there are no specifics as to whether the species was growing terrestrially or connected to wood. The original collection of *P. minutissimus* exists, but it is in poor condition and not available for study. However, curator Caroline Loup kindly revised the holotype. The conserved material consists of a single basidioma with a stipe 10 mm tall and a pileus 4 mm in diameter with only two visible pieces of protruding lamellae, less than 0.5 mm each. By her revision, basidiospores are globose, about 5–6 μm, a few clumps of cystidia are present, mostly clavate to (sub)fusiform; and basidia are collapsed. In the absence of modern collections from the type locality (Vallvidrera, Spain) or from similar habitats nearby, we are not in a position to accept or refute the synonymy of *P. podospileus* and *P. minutissimus*. It is possible that the taxon described by Kühner and Romagnesi as *P. minutissimus* f. *typicus* and by Orton as *P. minutissimus* is indeed a separate species from *P. podospileus* by virtue of the terrestrial habitat and the non-squamulose stipe. Whether *P. minutissimus* would be the correct name for such a species would have to be confirmed. In the present study, we recognize six species in the *P. podospileus* complex occurring in Europe, five of them molecularly confirmed from Spain or southern France: *Pluteus podospileus* (NBM-F-009808), *P. cutefractus* (BRNM825872, GM 3784, GM3458), *P. inflatus* (BRNM 825873), *Pluteus* sp. (GM3751), and *P. notabilis* var. *notabilis* (BRNM825867). Two of these species (*P. inflatus* and *Pluteus* sp. GM 3751) can have white, non-squamulose stipes and also grow terrestrially. *Pluteus* sp. GM 3751 differs from the original description of *Pluteus minutissimus,* at least in the strongly rugose-venose pileus. *Pluteus inflatus* is a geographically widespread species also confirmed to occur in Spain, and some collections of *P. inflatus* match the *P. minutissimus* protologue. It should be noted that *P. minutissimus* does not have priority over either *P. podospileus* or *P. inflatus*, the two species accepted here that more closely match its original description. However, we prefer to verify the taxonomic value of *P. minutissimus* strictly by molecular analysis of the holotype collection in the future when less invasive methods become available.



***Pluteus inflatus*** Velenovský Figure 4, Figure 5, Figure 6 and Figure 7.

České Houby **3**: 609 (1921) [20].

Syn.: *Pluteus inflatus* var. *alneus* Velen., Novitates Mycologicae: 145 (1939).

*Pluteus granulatus* var. *tenellus* J. Favre, Beiträge zur Kryptogamenflora der Schweiz 10 (3): 101 (Favre 1948).

*Pluteus minutissimus* f. *major* Kühner, Bull. Soc. Mycol. France 72 (3): 182 (1956); *Pluteus minutissimus* f. *major* Kühner, in Kühner and Romagnesi, Fl. Analyt. Champ. Supér. (Paris): 422 (1953). Nom. inval., Art. 39.1 (Shenzhen Code).

Holotype of *Pluteus inflatus* var. *alneus*: Czech Republic, Mnichovice, on trunk of *Alnus*, 7 August 1939, leg. J. Velenovský (PRM 154261).

Syntypes of *Pluteus minutissimus* f. *major*: France, Val de Marne, Bois de Vincennes, on a big stump near rivers, 21 September 1929, leg. and det. R. Kühner (G00126660; G00126659).

Holotype of *Pluteus granulatus* var. *tenellus*: Switzerland: Jura, haut-marais, tourbière du Bélieu pr. du Russey, Doubs, in association with *Filipendula ulmaria*, 15 August 1938 (G K9933, G00126179).

Pileus 10–30 mm in diameter, hemispherical or campanulate when young, expanding to convex to plano-convex, rarely plane, with or without indistinct, low and broad umbo, only rarely with distinct umbo, not or very rarely hygrophanous, smooth or slightly rugose at center, without striation or rarely translucently striate from margin up to half the radius of pileus, mostly not cracked, rarely with some cracks revealing the white context towards the margin, one collection cracked almost to the center. Surface dry, slightly to strongly velvety, becoming entirely minutely to distinctly granulose-squamulose (RAL 040 40 30), root brown (RAL 040 30 30), or wild brown (RAL 040 20 19), brown (S00 C00 Y60–S00 C00 Y70), red brown or chestnut brown, dark brown to almost back in the center (S00 C00 Y50). Lamellae free, crowded to moderately distant, mostly ventricose, up to 4 mm broad, white (Y10 M00 C00) or grey-white, later pinkish (Y10 M00 C00) to pink, with a concolorous, even, slightly pubescent to distinctly flocculose edge. Stipe 25–45(–60) × 1–2(–4) mm, cylindrical, sometimes slightly broadened at base, sometimes (finely) longitudinally fibrillose, whitish or greyish, entirely white pubescent or only in the lower half, sometimes with minute distinct brown scales in the lower part, tomentum at the base white. Context in stipe and pileus white or grey-white. Smell and taste indistinct.

Basidiospores [410/14/14] (4.5–)5.0–7(–7.7) × (4.0–)4.5–6(–6.5) µm, avl × avw = 5.3–6.2 × 4.8–5 µm, Q = (1.0–)1.1–1.3(–1.6); Q* = 1.13–1.30, subglobose to broadly ellipsoid, rarely globose or ovoid, thick-walled. Basidia 15–34 × 6–12 µm, 4-spored, rarely 2-spored, mostly clavate, less frequently almost cylindrical or subfusiform. Pleurocystidia absent or rare, rarely common all-over lamellar faces, 36–73(–80) × (10–)15–30 µm, narrowly clavate to broadly clavate or oviform, some ellipsoid, narrowly utriform or inflated-fusiform, hyaline, thin-walled. Lamellar edges heterogeneous and fertile, with cheilocystidia, basidia, and basidioles, with cheilocystidia crowded in places in clusters but not forming a continuous strip. Cheilocystidia 20–60(–65) × (10–)15–30 µm, numerous, narrowly to broadly clavate or cylindrical, ovoid or rarely narrowly utriform to utriform or subfusiform, hyaline, thin-walled. Pileipellis a hymeniderm, or rarely an epithelioid hymeniderm, made up of two types of elements: (i) spheropedunculate or broadly clavate to clavate, rarely subovoid 28–63(–66) × 16–42 µm; (ii) narrowly to broadly fusiform or cylindrical, rarely narrowly clavate to clavate or narrowly utriform, 72–130(–190) × (10–)16–30(–34) µm; all elements with pale brown to brown or yellow-brown intracellular pigment often aggregated into spots, thin to thick-walled. Stipitipellis a cutis of 5–12 µm wide cylindrical hyphae, slightly thick-walled, hyaline, rarely with pale brown pigment. Caulocystidia absent to numerous, often in clusters, narrowly to broadly fusiform, rarely clavate or flexuose, some with 2–3 basal septa, 40–145 × (7–)8–20 µm, colorless or with pale brown to brown, unevenly distributed intracellular pigment. Clamp connections absent in all studied tissues.

Habit, habitat, and phenology: mostly solitary, on very decayed wood of deciduous trees, on litter, or on soil. May–October.

Distribution: Eurasia and North America (Canada). Widespread in Europe (Czech Republic, Denmark, Russia, Slovenia, Sweden, Switzerland, UK—England) and Western Asia (Turkey, Iran).

Collections examined: EURASIA. **CZECH REPUBLIC:** Mirošovice, in the forest, August 1940, leg. J. Velenovský (PRM 154566); Mnichovice, Hrusice, *Crataegus*, 30 September 1940, leg. J. Velenovský (PRM 154565 without sequences, poor condition); Mnichovice, trunk of *Alnus*, August 1944, leg. and det. J. Velenovský (PRM 154569, originally as *P. inflatus* var. *alneus*); Ládví, between creek and pond, mixed floodplain forest (*Alnus*, *Betula*, *Quercus*, *Fagus*, *Populus tremula*, *Picea*), on soil between deciduous twigs, 25 August 2018, leg. J. Herčík (BRNM 817761); Olomouc, Černovírské slatiniště forest, about 220 m a.s.l., *Alnetorum* with *Ulmus minor* and *Quercus robur*, near wetland with *Typha*, wet deciduous wood, 26 August 2019, leg. V. Halasů (BRNM825839); Olomouc, ul. Václava III. Street 28, grass place in front of the house, in the grass, where the tree used to stand (probably *Fraxinus*), about 220 m a.s.l., 28 May 2019, leg. V. Halasů (BRNM825836); Štiřín, *Alnetorum* with *Quercus* and *Tilia*, on soil between deciduous twigs, 25 August 2018, leg. J. Herčík (BRNM825838); Valtice, Rendez-vous Nature Reserve, *Quercetum* with *Juglans nigra* on soil near footpath, 48°44,801′ 16°47,386′, 11 September 2016, leg. P. Včelička and V. Halasů (BRNM825837). **DENMARK:** Æbelø, on very rooty wood of *Alnus glutinosa* in deciduous forest, 1 September 2021, leg T. Læssøe and M. Schier Christiansen, DMS-10206994 (C); Brobæk Mose at Lyngby, N. of Copenhagen, on dead wood of *Betula* in bog woodland, 22 September 2016, leg T. Kehlet, DMS-10027661 (C); E Jutland, Havskov N. of Aarhus, on rotten wood of *Fagus sylvatica* in deciduous forest, 30 August 2021, leg. U. Nygaard and T. Kehlet, DMS-10205646 (C); North Zealand, Glostrup, 30 September 2019, leg. Thomas Kehlet, on soil on roadside in suburban area, DMS-10041511 (C). **IRAN:** Golestan, Gorgon, more or less virgin beech forest at 950 m elevation, on rotten log of *Fagus orientalis*, 3 October 2016, leg. J. Heilmann–Clausen and C. Bässler, DMS-10334330 (C). **RUSSIA:** Leningrad Region, Saint Petersburg, Botanical Garden, 22 September 1946, leg. B.P. Vassilkov (LE F-9809); Orenburg Region, Belyaevsky District, Orenburg Nature Reserve, site «Burtinskaya Steppe», *Alnus glutinosa* spinney near Kainar stream, on soil and woody debris among moss, 22 July 2004, leg. O.A. Desyatova (LE F-235208); Caucasus, Karachayevo-Circassian Republic, Teberda Biosphere Reserve, vicinity of Teberda, Dzhemagat Gorge, mixed forest (*Pinus*, *Betula*, *Populus*), on soil among moss or on very decayed deciduous wood, 15 August 2009, leg. E.F. Malysheva (LE F-262712). **SLOVENIA:** Nova Goricâ, Soča Park, 16 September 2018, leg. G. Ferisin (FG16092018026); ibid., 25 June 2017 (FG 250620179). **SPAIN:** Cataluña, Villalonga de Ter, Abella, in a humid valley, in a mixed forest, under *Corylus avellana*, *Quercus pyrenaica*, *Fagus sylvatica,* etc., on muddy ground, 1200 m, 30 August 2022, leg. S. Gibert, À. Torrent and G. Muñoz (GM3991); Espinavell, Molló (city), in riparian forest with *Alnus glutinosa*, *Corylus avellana*, *Fagus sylvatica*, and *Betula pendula*, 1400 m a.s.l., on acid soil, 15 July 2021, leg. S. Gibert (det. as *P. minutissimus* BRNM 825873). WESTERN ASIA. **TURKEY.** Artvin Province, Borçka Region, Camili Biosphere Reserve Area, on broken pieces of trunks and branches of *Fagus orientalis*, 1485 m a.s.l., 2 October 2011, leg. O. Kaygusuz (OKA-TR260); Rize Province, Çamlihemşin Region, near Ayder Upland, on decayed wood of *F. orientalis*, 1250 m a.s.l., 28 September 2012, leg. O. Kaygusuz (OKA-TR261). UNITED KINDGDOM. **ENGLAND:** London, Kew Botanical Gardens, terrestrial, under *Celtis occidentalis*, next to a cluster of *Psathyrella lacrymabunda*, 21 July 2009, leg. G. Muñoz (GM1524). NORTH AMERICA. **CANADA:** Québec, Saint-Marc-des-Carrières, 48 m a.s.l., at the edge of a parking lot surrounded by large *Picea* sp., in rocky-sandy soil, 28 June 2022, leg. R. Lebeuf (HRL3689).

Notes: Collection PRM 154569 was labeled as a holotype of *P. inflatus* var. *alneus* on a fungarium sheet but was collected after the publication of the protologue. The curator of PRM, J. Holec, confirmed (pers. comm.) that this collection was not labeled as a holotype by Velenovský but was labeled as such by a later worker. The original material of *Pluteus inflatus* mentioned in the protologue was also collected by Velenovský from the village of Mnichovice on an *Alnus* trunk, but earlier (7 August 1939, PRM 154261). This collection is the only material that perfectly matches the protologue. Thus, we recognize PRM 154261 to be the true holotype. The holotype of *P. inflatus* does not exist. The holotype of *Pluteus inflatus* var. *alneus* (PRM 154261) as well as PRM154569 and *Pluteus inflatus* PRM154566 fall into the same/*inflatus* clade, while one collection (PRM154563 as *Pluteus inflatus*) falls into the/*cutefractus* clade. Velenovský [20] described *P. inflatus* var. *inflatus* with a non-cracked pileus without striation, growing on deciduous (*Carpinus*) stumps. This description confirms the/*inflatus* clade, supported by three collections, as the best candidate for the original concept of *P. inflatus*. Velenovský [21] described a variety of *Pluteus inflatus* var. *alneus* only with the short note that it grows on the trunk of *Alnus* and that it is easily recognized. Because species of the genus *Pluteus* often grow on substrates of various deciduous trees (e.g., [1,2,7,53]), we do not believe that growth on *Alnus* versus *Carpinus* is sufficient to describe a separate variety. Thus, we synonymize *Pluteus inflatus* var. *alneus* under *Pluteus inflatus*.

Both sequenced syntypes of *Pluteus minutissimus* f. *major* (G00126660 and G00126659) belong to *P. inflatus*, which has priority. Thus, *Pluteus minutissimus* f. *major* is only a later synonym of *P. inflatus*. Surprisingly, our molecular studies show that the holotype of *Pluteus granulatus* var. *tenellus* J. Favre (G K9933, G00126179) also represents *P. inflatus*, even though Favre [54] described this variety as having a pileipellis composed of three types of elements, while there are only two significant element types in a typical basidioma of *P. inflatus*. Unfortunately, the size and condition of the *Pluteus granulatus* var. *tenellus* holotype do not allow a precise verification of the pileipellis structure in numerous parts of the pileipellis, but, based on our studies, there are various elements that can be interpreted as representing two or three types by their similarities. *Pluteus inflatus* is a highly variable species that is difficult to distinguish from related species and has a broad ecology, being recorded from both soil, plant litter, and decaying wood.

*Pluteus inflatus* is widely distributed in Eurasia, and even two North American collections have been molecularly confirmed. Both collections were made in disturbed, man-made habitats (urban gardens, parking lots), which could indicate a recent, human-mediated introduction to North America, but more collections are needed to confirm this hypothesis.



***Pluteus absconditus*** Justo, Kalichman and S.D. Russell **sp. nov.**
Figure 8 and Figure 9.

MB 849337.

Etymology: *absconditus*, meaning hidden or concealed, because of the difficulty separating this species from *P. inflatus* without molecular data.

*Diagnosis*: Closely related to *P. inflatus*, differing in the more abundant pleurocystidia, shorter cheilocystidia, and different nrITS (5–8 evolutionary changes) and *TEF1-α* sequences (5 evolutionary changes).

*Holotype*: USA, Tennessee: Loudon Co., Greenback, East Lakeshore Trails, on soil in mixed forest with *Juniperus virginiana*, *Pinus*, and hardwoods (*Acer*, *Quercus*, *Cornus florida*, *Juglans nigra*, etc.), 15 June 2013, leg. C. Braaten, Mushroom Observer 136488 (NBM-F-009787).

Pileus 15–40 mm in diameter, hemispherical when young, expanding to convex or plano-convex, with a low, broad umbo; surface strongly granulose-squamulose all over, with cracks that reveal the white context underneath, sometimes with a pulverulent-powdery aspect and/or with white-translucent pubescence; rugulose at center in young specimens; with predominant brown to reddish brown colors (Munsell: 5YR 4/6, 5/6–5/8, 6/6–6/8; 2.5YR 4/8, 5/8), distinctly darker at center; dry, not hygrophanous; margin rimose or (translucently) striate. Lamellae crowded, free, ventricose, up to 5 mm broad, white when young, later pink, with white or concolorous, flocculose, or smooth edges. Stipe 25–45 × 2–4 mm, cylindrical, with a slightly broadened base; surface white, silvery-white, shiny, or sometimes with a translucent aspect; pubescent-hairy all over, at least in young specimens; hairs sometimes grouped in mats with pale brown tips. Context in stipe and pileus white. Smell and taste none.

Basidiospores [60/2/2] 5.0–6.8 × 4.3–5.8 μm, avl × avw = 5.7–6.0 × 4.7–5.3 μm, Q = 1.00–1.26, Q* = 1.13–1.23, globose to broadly ellipsoid. Basidia 16–27 × 6.5–9 μm, predominantly 4-spored, but 2-spored and 1-spored basidia are also relatively common, clavate. Pleurocystidia 41–66 × 18–22(–26) μm, mostly narrowly clavate, ovoid, or ellipsoid; hyaline; very rarely with pale-brown intracellular pigment; thin-walled; common to scattered, all over lamellar faces. Lamellar edges heterogeneous, with cheilocystidia, basidia, and basidioles. Cheilocystidia 22–42(–49) × 11–22 μm, mostly (narrowly) clavate, subglobose, or spheropedunculate, rarely ovoid, hyaline, thin-walled, scattered, or in clusters, not forming a well-developed strip. Pileipellis a hymeniderm made up of two types of elements: (i) spheropedunculate or (narrowly) clavate, narrowly utriform or fusiform, 36–61 × 19–42 µm; (ii) fusiform, narrowly utriform, cylindrical, 75–127 × 19–28 µm; all elements with brown intracellular pigment, slightly thick-walled. Stipitipellis a cutis of cylindrical, slightly thick-walled, 6–10 µm wide hyphae, hyaline, or rarely with pale brown pigment. Caulocystidia common, often in loosely arranged clusters, 36–125 × 6–25 µm, cylindrical, narrowly clavate, narrowly fusiform, sometimes with a flexuous and/or rounded apex; often with 2–3 shortly-septate basal elements, 14–24 × 9–15 µm; hyaline or with pale brown intracellular pigment, sometimes with parietal pigment at the apex. Clamp connections absent in all studied tissues.

Habit, habitat, and phenology: solitary to subgregarious, growing on soil, often among leaves, twigs, and other plant matter, but without apparent connection to wood. June-August.

Distribution: Eastern North America: known from the USA (Indiana, Tennessee).

Additional collections examined: **USA. Indiana:** Monroe Co., Bloomington, Griffy Woods Nature Preserve, among woody debris, 28 August 2018, leg. Ron Kerner, iNaturalist 16154325; Brown Co., Nashville, 29 July 2022, leg. Brian Hunt, iNaturalist 128466549; ibid., 5 August 2022, iNaturalist 129614045. **Tennessee:** Anderson Co., Oak Ridge, Haw Ridge Park, 20 July 2019, leg. J. Kalichman, iNaturalist 112240775 (NBM-F-009786).

Notes: *Pluteus absconditus* is recovered in the phylogenetic analysis as a sister to *Pluteus inflatus*. Their nrITS sequences differ in 5–8 evolutionary changes, and their *TEF1-α* sequences in 5. Morphologically, both species are very similar, with *P. inflatus* differing in the rare or absent pleurocystidia and the more abundant and slightly larger cheilocystidia (20–60(–65) × (10–)15–30 µm). *P. absconditus* has 2-spored and 1-spored basidia, which is not a common character for *Pluteus* species, where most taxa have exclusively 4-spored basidia. The pileus of *Pluteus absconditus* seems to be paler than typical *P. inflatus*; however, the latter is an extremely variable species, and the variability of *P. absconditus* has not been well investigated yet.



***Pluteus cutefractus*** Ferisin, Dovana, and Justo Sydowia 71: 285 (2020); Ferisin et al. [27] Figure 10 and Figure 11.

*Pluteus cutefractus* Ferisin, Dovana, and Justo Sydowia 71: 182 (2019), in Song et al. [55], Nom. inval.

Original diagnosis. Basidiomata with a markedly cracked pileipellis, globose to broadly ellipsoid basidiospores, predominantly ovoid to broadly clavate cheilocystidia and pleurocystidia, and trichohymeniderm pileipellis consisting of broadly utriform and fusiform elements.

General description: Pileus 10–29 mm broad, plano-convex when young, later plane, plano-concave to concave, not hygrophanous, venose or smooth in center, striate at margin, often cracked from margin up to 2/4–3/4 of pileus, more conspicuously when mature, showing whitish flesh underneath; fawn brown (RAL 8007), nut brown (RAL 8011), beige brown (RAL 8024), darker in the center. Lamellae free, moderately crowded, slightly ventricose, up to 4 mm broad, whitish when young, later light to dirty pink, sometimes with a yellowish tinge, with a whitish or concolorous flocculose edge. Stipe 29–51 × 2–5 mm, cylindrical, not bulbous, but sometimes 0.5–2 mm broadened, pubescent, sometimes finely longitudinally fibrillose, whitish or hyaline grayish, sometimes squamulose to densely covered with distinct rusty brown to brown floccules more conspicuous (more closely covered) in the lower part. Context whitish. Smell and taste indistinct.

Basidiopores [360/14/14] 4.9–7.1(–7.3) × (4.2–)4.5–5.8(–6.4) μm; avl × avw = 5.7–6.0 × 5–5.4 µm; Q = (1.00–)1.04–1.20(–1.33); Q* = 1.12–1.14, globose to broadly ellipsoid, thick-walled, inamyloid. Basidia 21–30 × 8–10 μm, clavate, 4-spored, very rarely 2-spored. Pleurocystidia (33–)40–55 × 20–25(–26) μm, scattered, ovoid, clavate to broadly clavate, rarely subfusiform with an obtuse apex, thin-walled, hyaline. Lamellar edges sterile, rarely heterogeneous. Cheilocystidia (18–)26–76 × 12–26 μm, ovoid, clavate to broadly clavate, rarely broadly subfusiform with obtuse apex, thin-walled, hyaline. Pileipellis a hymeniderm made up of broadly utriform (45–60 × 20–35 μm) and narrowly fusiform (100–150 × 11–35 μm) elements; pigment intracellular, vacuolar, light brown or brown. Stipitipellis a cutis of whitish hyphae 4–10(–12) μm wide. Caulocystidia present, descending to about 1/3 of the length of the stipe (36–)40–60 × 12–18 μm, variable in shape from clavate to fusiform, sometimes filled with evenly dissolved brownish intracellular pigment. Clamp connections absent in all studied tissues.

Habit, habitat, and phenology: solitary or in groups, on soil or on broadleaved trunks and stumps, from summer to autumn.

Distribution: known only from Europe (Czech Republic, Denmark, France, Slovenia, Spain).

Holotype: Slovenia: Nova Goricâ, Panoveč Park, with *Abies alba* and *Fagus*, on deciduous stump, 8 July 2017, leg. G. Ferisin (MCVE 30110).

Additional collections examined: **CZECH REPUBLIC**: Mnichovice, Jidášky, on a fallen trunk of *Betula*, September 1940, leg. J. Velenovský (PRM 154563 as *Pluteus inflatus*); Grygov, Království forest, under *Alnus glutinosa*, *Fraxinus excelsior*, *Tilia*, *Quercus*, *Padus racemosa*, on soil and deciduous stump, 23 June 2019, leg. V. Halasů and H. Ševčíková (BRNM 816221). **DENMARK**: Northen Zealand, Højbjerg Hegn, on soil in deciduous forest, leg. M. Sonniks, DMS-10213877 (C). **FRANCE**: Oise, in a muddy rut of a trail, deciduous forest, 3 August 1980, leg. and det. H. Romagnesi (MNHN-PC-FUSION111827, originally as *Pluteus minutissimus*). **SLOVENIA:** Harije, under *Fagus*, on deciduous stump, 17 July 2019, leg. G. Ferisin (FG 17072019); Nova Goricâ, Panoveč Park, with *Abies alba* and *Fagus*, on deciduous stump, 26 September 2015, leg. G. Ferisin (FG 26092015); ibid., 8 July 2017 (MCVE 30111); ibid., 8 July 2017 (FG12794); ibid., 2 June 2018, leg. G. Ferisin (MCVE 30143), 9 September 2018 (FG 09092018). **SPAIN**: Fitor (city), in riparian forest with *Corylus avellana* and *Alnus glutinosa* (also *Quercus* spp.?), on acid soil, 400 m a.s.l., 7 June 2008, leg. C. Roqué (det. as *P. podospileus* by C. Roqué, BRNM 825872); Navarra, Oieregi, Señorío de Bértiz, on the sand deposited at the root of a completely fallen broadleaved unidentified tree, among many *Coprinellus disseminatus*, in a riverside forest with abundant allochthonous vegetation (*Platanus* x *hispanica*, *Quercus rubra*, *Trachycarpus fortune*, *Prunus laurocerasus*, *Diospyros virginiana*, *Phyllostachys viridiglaucescens*, etc.), 25 September 2021, leg. G. Muñoz (GM3784); Cataluña, Sant Hilari Sacalm, under *Alnus glutinosa*, *Corylus avellana* and *Fraxinus excelsior*, apparently on soil, 14 September 2019, leg. S. Gibert (GM3458, duplo in SGS20190914.10).

Notes: *Pluteus cutefractus* was described as having a markedly cracked pileus margin by Song et al. [55]. This feature can be relatively indistinct in some young basidiomata. Moreover, many other species may have cracked pileus in extreme weather conditions. Thus, this feature must be considered carefully, and microscopic features and ecology must be taken into account.



***Pluteus fuscodiscus*** Ferisin, E.F. Malysheva, Ševčíková, Kaygusuz, Heilm.-Claus. and I. Saar **sp. nov.**
Figure 12, Figure 13, Figure 14 and Figure 15.

MB 849803.

*Etymology*: *fuscodiscus* refers to the dark central disc on the pileus.

*Diagnosis*: Differs from other known species of the *Pluteus podospileus* complex by having a pileus with a distinctly darker disk in the center.

*Holotype:* Italy: Friuli Venezia Giulia, Farra d’Isonzo, and Bosco di Sotto, in small groups, on twigs of broadleaved trees (*Salix*, *Quercus,* and *Populus*), 22 August 2015, leg. G. Ferisin (FG22082015).

Pileus 5–30 mm in diameter, campanulate or convex to plano-convex, then applanate, with indistinct umbo, not or slightly hygrophanous, indistinctly striate near the pileus margin to striate up to half of radius, margin slightly eroded; surface granulose-pruinose to granulose-squamulose, velvety, cracking at the margin to show underlying whitish context or not, darkly colored night brown (RAL 050 20 16) or granite brown (RAL 050 20 10) when young, then golden brown (RAL 050 50 30), brazilian brown (RAL 060 40 309), coffee bean brown (RAL 060 40 20), madeira brown (RAL 050 40 40), chalk beige (RAL 075 80 10) to chestnut brown (RAL 040 40 30), with a distinctly darker disk in the centre: mahogany brown, chocolate brown, terra brown or darker, slightly paler towards the margin. Lamellae free, crowded to moderately crowded, ventricose, whitish, later pinkish, with concolorous flocculose eroded edges. Stipe 12–45 × 1–4 mm, cylindrical or faintly thickened downwards, often with a slightly broadened base; surface white, whitish pellucid, or greyish brown, at least on the lower half covered with brown to almost black floccules, often more densely at the base. Context whitish. Smell indistinct; taste not recorded.

Basidiospores [435/13/13] (4.5–)4.8–6.5(–6.8) × (3.46–)4.0–5.5(–5.9) µm, avl × avw = 5.3–5.7 × 4.2–4.8 μm, Q (1.0–)1.10–1.36(–1.47), Q* = 1.10–1.26, subglobose to ellipsoid, rarely globose, thick-walled. Basidia 16–32 × 6–12 µm, 4-spored. Pleurocystidia absent or very rare, 34–36 × 10–13 µm, utriform to spathulate, hyaline, thin-walled. Lamellar edge sterile. Cheilocystidia 25–76 × 10–30(–55) µm, clavate to almost cylindrical, broadly clavate, oviform, broadly utriform, in some collections frequently mucronate, hyaline, or with light brown intracellular pigment, rarely with oily contents, thin-walled. Pileipellis a hymeniderm made up of two types of elements: (i) spheropedunculate or broadly clavate, (30–)37–62 × 21–55 µm; (ii) (narrowly) fusiform, inflated-fusiform, sometimes mucronate, (57–)63–180 × (10–)12.5–33 µm; all elements with light to dark brown or yellow-brown intracellular pigment, rarely with oily contents, thick-walled. In some collections, there are not two types of elements but rather variable short-to-long elements. Stipitipellis a cutis of cylindrical, hyaline, thin-walled to slightly thick-walled hyphae, 6–10 µm wide. Caulocystidia numerous or rare, throughout full length, in tufts or solitary, 30–143 × 6–22(–24) µm, cylindrical, narrowly to narrowly fusiform, broadly clavate or broadly utriform, with yellow-brown to brown intracellular pigment, in some collections often aggregated into spots. Clamp connections absent in all studied tissues.

Habit, habitat, and phenology: solitary or in small groups, growing on decayed deciduous or coniferous wood, sawdust, or other plant litter. In July–October.

Distribution: Europe: Czech Republic, Estonia, France, Italy, European part of Russia; and Asia: Iran, Turkey, Far East Russia.

Additional collections examined: EUROPE. **CZECH REPUBLIC:** Lanžhot, Ranšpurk Nature Reserve, floodplain forest, fallen deciduous trunk, 3 October 2014, leg. H. Ševčíková (BRNM 766589); ibid., 15 October 2019, leg. H. Ševčíková (BRNM 817683). **FRANCE:** Saint-Gatien-des-Bois, Calvados, *Fagus*, 13 July 2017, leg. T. Duchemin (BRNM 825871). **ESTONIA:** Lääne Co., Taebla Comm., Palivere skiing track, 58.96537° N, 23.90702° E, on sawdust, 27 September 2012, leg. V. Liiv (TUF118497). **ITALY:** Friuli Venezia Giulia, Farra d’Isonzo, 2 August 2015, leg. G. Ferisin (DOV 653, FG 02082015007); ibid., 20 August 2015, leg G. Ferisin (FG 12152); ibid., 11 August 2017, leg. G. Ferisin (ALV 12977-FG 11082017). **RUSSIA: European part**, Samara Region, Zhiguli Biosphere Reserve, vicinity of Bakhilova Polyana, birch forest, on decayed wood of deciduous trees, 15 August 2004, leg. E.F. Malysheva (LE F-313448). ASIA. **RUSSIA: Far East**, Primorye Territory, Sikhote-Alin Nature Biosphere Reserve, vicinity of Ust’-Shandui Reserve Field Station, plateau near Zabolochennaya river, mixed forest with *Tilia* sp. and *Pinus koraiensis*, on wood of coniferous tree, 23 August 2012, leg. E.F. Malysheva (LE F-313336); Primorye Territory, Sikhote-Alin Nature Biosphere Reserve, vicinity of Yasny Reserve Field Station, floodplain of Yasnaya river, mixed forest with *Pinus koraiensis*, *Tilia* sp., *Quercus mongolica*, *Populus maximowiczii*, on litter in fern thickets, 26 August 2013, leg. O.V. Morozova (LE F-312862). WESTERN ASIA. **IRAN:** Golestan, Gorgon, +/−virgin beech forest, 950 m a.s.l., on rotten log of *Fagus orientalis*, 3 October 2016, leg. Jacob Heilmann-Clausen and Claus Bässler (DMS-10334331). **TURKEY:** Muğla Province, Köycegiz District, near Dögüşbelen town, on decayed wood of *Liquidambar orientalis*, 6 m a.s.l., 18 November 2010, leg. O. Kaygusuz (OKA-TR691); ibid., on decayed wood of *L. orientalis*, 10 m a.s.l., 16 December 2010, leg. O. Kaygusuz (OKA-TR695); ibid., on rotten wood of *L. orientalis*, 9 m a.s.l., 22 November 2011, leg. O. Kaygusuz (OKA-TR715); ibid., on fallen trunk or well-decayed wood of *L. orientalis*, 5 m a.s.l., 18 March 2012, leg. O. Kaygusuz (OKA-TR749); ibid., on well-decayed wood of *L. orientalis*, 8 m a.s.l., 25 March 2012, leg. O. Kaygusuz (OKA-TR750).

Notes: Typical basidiomata of *Pluteus fuscodiscus* have a granulose-pruinose, velvety pileus with a distinctly darker disk in the center, a white to greyish brown stipe with at least the lower half covered with brown to almost black floccules, a pileipellis made up of two types of elements, and an absence of utriform to spathulate pleurocystidia.



***Pluteus seticeps*** (G.F. Atk.) Singer Figure 16 and Figure 17.

Lloydia **21**(4): 272 (1959)1958.

Basionym: *Leptonia seticeps* G.F. Atk., J. Mycol. 8(3): 116 (1902).

Syn.: *Leptoniella seticeps* (G.F. Atk.) Murrill, North American Flora 10: 92. 1917.

*Pluteus psychiophorus* var. *seticeps* (G.F. Atk.) Singer, Trans. Br. Mycol. Soc. 39(2): 214 (1956) (nom. inval; published as “ad. int.” Art. 36.1).

*Pluteus nanellus* Murrill, N. Amer. Fl. (New York) 10(2): 130 (1917) (see Minnis and Sundberg 2010) [13].

*Missapl. Pluteus podospileus* Sacc. and Cub., Syll. fung. (Abellini) 5: 672 (1887).

*Excluded. Pluteus seticeps* sensu Homola (1972) (probably *Pluteus cystidiosus*).

Pileus 8–24 mm in diameter, hemispherical or campanulate when young, expanding to convex or plano-convex, often with a low, broad umbo; surface soon becoming granulose-squamulose all over, often becoming minutely rimulose with age, showing the white context underneath, especially towards the margin, with the center remaining smooth or rugose; with predominant brown colors (Munsell: 5YR 4/6, 5/6–5/8, 6/6–6/8; 7.5YR 4/6, 5/6–5/8, 6/6–6/8), darker at center; dry, not hygrophanous; margin often translucently striate up to half the radius of pileus, often rimose. Lamellae crowded, free, ventricose, up to 4 mm broad, white when young, later pink, with even, serrated, or flocculose edges, white or concolorous. Stipe 15–28 × 1–2 mm, cylindrical, with slightly broadened base; surface white, shiny, sometimes with grey, grey-brown hues at base; surface smooth or with some appressed brown fibrils in the lower part of the stipe. Context in stipe and pileus white. Smell and taste none.

Basidiospores [200/11/9] 4.0–6.0(–6.5) × (3.5–)4.0–5.5 μm, avl × avw = 4.6–5.0 × 4.3–4.8 μm, Q = 1.00–1.26, Q* = 1.03–1.13, globose to broadly ellipsoid, some ovoid. Basidia 18–26 × 6.5–8 μm, 4-spored, clavate. Pleurocystidia absent from lamellar faces; very rarely, some cystidia present near the lamellar edge. Cheilocystidia 25–56(–91) × 10–30 μm, mostly clavate, narrowly clavate or ellipsoid, some narrowly utriform or ovoid; hyaline, thin-walled, crowded, forming a well-developed strip or in discrete clusters, making the lamellar edge heterogeneous. Pileipellis a hymeniderm or epithelioid hymeniderm, made up of two types of elements: (i) spheropedunculate, pyriform, or broadly clavate, 30–69 × 13–38 µm; (ii) fusiform, lageniform, narrowly utriform, often mucronate, (70–)80–111 × 11–27 µm; all elements with brown intracellular pigment, evenly dissolved or aggregated in spots, slightly thick-walled. Stipitipellis a cutis of cylindrical, slightly thick-walled, 6–10 µm wide hyphae; hyaline, or some with pale brown pigment; at the base of the stipe with groups of slightly irregular, adnate hyphae, heavily pigmented. Caulocystidia scattered, mostly not occurring in clusters, 44–89 × 13.5–27 µm, narrowly utriform, lageniform, fusiform, clavate, some mucronate; most with brown pigment, often densely aggregated. Clamp connections absent in all studied tissues.

Habit, habitat, and phenology: solitary or gregarious, growing on well-decayed wood of angiosperms (e.g., *Acer*). In temperate or transitional boreal/temperate forests. July–August. According to Minnis and Sundberg [13], also June and September.

Distribution: North America: widely distributed in eastern North America. Molecularly confirmed from Canada (Québec) and the USA (Illinois, Indiana, Missouri, New Jersey, Ohio, Pennsylvania, Tennessee, and Wisconsin).

Collections examined: **CANADA. Québec:** Saint-Laurent, on decayed angiosperm wood, 5 July 2011, leg. Renée Lebeuf, HRL0715 (NBM-F-009796); ibid., 7 July 2011, HRL0716 (NBM-F-009797). **USA. New Jersey**: Gloucester County, Mantua, Chestnut Branch Park, on decayed hardwood, 8 August 2021, leg. M. Patiño, iNaturalist 91066075 (NBM-F-009802); Morris County, Mendham, Meadowood Park, on decayed hardwood, 24 July 2021, leg. M. Patiño, iNaturalist 88771421 (NBM-F-009801). **Ohio:** Franklin County, Gahanna, on a well-rotted, mossy log in floodplain near *Platanus occidentalis*, *Populus deltoides*, *Asimina triloba*, *Acer negundo,* and *Aesculus flava*, 17 July 2014, leg. D. Grootmyers, MO 177404 (NBM-F-009809). **Pennsylvania:** Luzerne County, Sweet Valley, on a pile of wood chips and other woody debris, 27 July 2018, leg. D. Wasilewski, MO 324271 (NBM-F-009810). **Tennessee:** Anderson Co., Oak Ridge, Haw Ridge Park, 14 June 2019, leg. J. Kalichman, iNaturalist 112235090 (NBM-F-009798); ibid., 23 June 2019, iNaturalist 112237607 (NBM-F-009799); ibid., 1 July 2021, iNaturalist 112335052 (NBM-F-009800).

Notes: *Pluteus seticeps* has been considered by some authors to be a synonym of *P. podospileus* [2,4], but detailed revision of the type collection, modern North American collections, and molecular data clearly indicate these are separate species [7,13]. Both species have non-overlapping geographical ranges, with *P. podospileus* occurring only in Eurasia and *P. seticeps* only present in eastern North America. Minnis and Sundberg [13] confirmed the usage of the name *Pluteus seticeps* for North American collections lacking pleurocystidia and established *Pluteus seticeps* var. *cystidiosus* Minnis and Sundb. for North American collections similar to *P. seticeps* with pleurocystidia. Molecular data from the type collection of *P. seticeps* var. *cystidiosus* indicate that this taxon is not closely related to *P. seticeps* but instead represents a sister taxon to the European *P. podospileus*. The new combination *P. cystidiosus* is adopted here for this species. Morphologically, *P. seticeps* can be distinguished from *P. cystidiosus* by the smaller basidiomata, pileus surface more markedly rimulose, stipe surface not covered in brown squamules, absence of pleurocystidia, and smaller caulocystidia that mostly do not occur in clusters.

*Pluteus nanellus* Murrill was described from New York as a species with small basidiomata, pale bay (pale reddish brown), minutely tomentose (visible only under the lens) pileus turning castaneous on drying, and a snow-white stipe [25]. Singer [22] and Homola [24,56] synonymized *Pluteus nanellus* and *P. seticeps* mostly based on the similarity of the pileipellis elements. Minnis and Sundberg [13] revised the type collections of both taxa and considered them to be identical. The only significant difference in the original descriptions of these species is the presence of some brown fibrils at the base of the stipe of *P. seticeps*, while this character is not mentioned in *P. nanellus*. Unfortunately, stipes are absent in both types of collections. Many of the specimens examined here have some brown fibrils at the base, but completely glabrous stipes also occur. We agree with Minnis and Sundberg [13] and consider *P. seticeps* and *P. nanellus* synonymous.



***Pluteus cystidiosus*** (Minnis and Sundberg) Justo, Malysheva and Lebeuf **comb. & stat. nov.** Figure 18 and Figure 19.

MB 849340.

Basionym: *Pluteus seticeps* var. *cystidiosus* Minnis and Sundberg N. Amer. Fung. 5(1): 13 (2010).

Holotype: USA: Michigan, Chippewa Co., White House Landing, on deciduous wood, 5 July 1965, leg. R. Homola, coll. RL Homola 1340 (MICH 69572).

Pileus 14–35(–40) mm in diameter, hemispherical or campanulate when young, expanding to convex or plano-convex, often with a low, broad umbo; surface pruinose, slightly to strongly velvety, becoming granulose-squamulose especially around center, smooth or strongly rugose all over when young, in older specimens smooth or rugose at center, for the most part not cracked, or with some cracks revealing the white context towards the margin, especially in older specimens; with predominant brown or red-brown colors (Munsell: 2.5YR 4/6–4/8, 3/4–3/6; 5YR 5/4–5/8, 4/4–4/6; when young 5YR 3/3, 2.5/2; RAL color chart: chestnut brown (RAL 040 40 30), red brown (RAL 020 20 20) or wild brown (RAL 040 20 19), usually darker at center; dry, hygrophanous; margin entire, serrate, slightly rimose or translucently striate. Lamellae crowded, free, ventricose, up to 5 mm broad, white when young, later pink, with even, serrated, or flocculose edges, white or concolorous. Stipe 15–40 × 1–5 mm, cylindrical, with a slightly broadened base; surface white, entirely minutely squamulose with dark or black-brown squamules, more densely so in the lower half. Context in stipe and pileus white. Smell and taste none.

Basidiospores [210/7/7] 4.5–5.5(–6.2) × 3.5–5.0 μm, avl × avw = 5.1–5.3 × 4.2–4.6 μm, Q = (1.00–)1.07–1.30, Q* = 1.11–1.24, subglobose to broadly ellipsoid, rarely globose or ovoid. Basidia 18.5–26 × 5.5–9 μm, 4-spored, clavate. Pleurocystidia 35–73(–82) × 11–31 μm, mostly (narrowly) utriform, also ovoid, clavate, or (broadly) fusiform; hyaline, thin-walled; scattered or rare over lamellar faces; rarely absent. Lamellar edge sterile. Cheilocystidia 26–63(–78) × 10–22(–26) μm, narrowly to broadly clavate, (narrowly) utriform or ovoid; hyaline, thin-walled, crowded, forming a well-developed strip. Pileipellis a hymeniderm or epithelioid hymeniderm, made up of two types of elements: (i) spheropedunculate, pyriform, or broadly clavate (30–)36.5–70 × 20–32 µm; (ii) broadly fusiform, inflated-fusiform, lanceolate, narrowly utriform, often mucronate, 66–121.5 × 9–17(–32) µm; all elements with brown or yellow-brown intracellular pigment, often aggregated in spots, slightly thick-walled. Stipitipellis a cutis of cylindrical, slightly thick-walled, 7–10 µm wide hyphae, hyaline, or some with pale brown pigment. Caulocystidia common, often in clusters, 36–125 × 6–25 µm, cylindrical, narrowly clavate, narrowly fusiform, spheropedunculate, often septate; most with brown or yellow-brown pigment, sometimes aggregated. Clamp connections absent in all studied tissues.

Habit, habitat, and phenology: often gregarious, growing on well-decayed wood of angiosperms (e.g., *Acer*, *Betula*), also on still-standing trees. In temperate or transitional boreal/temperate forests. July–October.

Distribution: North America: molecularly confirmed from Canada (New Brunswick, Québec) and the USA (Massachusetts, Michigan, Minnesota, and New York). Eurasia: molecularly confirmed from Japan and the Russian Far East (Primorye Territory).

Additional collections examined: NORTH AMERICA. **CANADA**. **New Brunswick**: Albert County, Caledonia Gorge Protected Natural Area, 6 km NNW of Riverside Albert, where Crooked Creek meets Caledonia Brook, 45.79731, -64.77409, on decayed wood, 6 July 2011, leg. E. Duke, EKD2011-91 (NBM-F 07400). **Québec:** Saint-Laurent, on angiosperm wood, 5 July 2011, leg. R. Lebeuf, HRL0713 (NBM-F-009792); Saint-Narcisse, Parc de la rivière Batiscan, Secteur Barrage, trail La P’tite vite, mixed forest dominated by *Tsuga canadensis*, *Fagus grandifolia,* and *Pinus strobus*, on the lying trunk of *Betula*, 16 July 2019, leg. R. Lebeuf and André Paul, HRL2941 (NBM-F-009791). **USA**. **Massachusetts:** Berkshire County, Savoy Mountain State Forest, at the base of a standing maple tree, 4 October 2012, leg. A. Justo, AJ 782 (NBM-F-009790). **New York**; Essex Co., Adirondack Ecological Center, Huntington Wildlife Forest, mixed forest, on decayed angiosperm wood, 15 August 2012, leg. A. Justo, AJ 617 (NBM-F-009788); ibid., 14 August 2012, AJ 619 (NBM-F-009789). EURASIA. **JAPAN**. Hokkaido Pref., Shiribeshi Prov., Otaru-shi, Nagahashi, on decayed broad-leaved wood, 17 July 2005, leg. K. Yamamoto (TNS-F-12398). **RUSSIA**: **Far East**, Primorye Territory, Sikhote–Alin Biosphere Reserve, vicinity of Yasny Reserve Field Station, floodplain of Yasnaya river, *Larix cajanderi* forest with *Betula platyphylla*, on rotten trunk of *Betula*, 14 August 2012, E.F. Malysheva (LE F-312852); Primorye Territory, Land of the Leopard National Park, plateau near Ananjevka river, mixed forest (*Quercus mongolica*, *Carpinus cordata*, *Ulmus japonica*, *Abies holophylla*), on mossy trunk of deciduous tree, 1 September 2011, leg. A.E. Kovalenko (LE F-313335).

Notes: *Pluteus cystidiosus* is recovered in the phylogenies as the sister species of *Pluteus podospileus*, and both species show an almost total overlap in their morphological variation, with *P. cystidiosus* having slightly larger pleurocystidia (35–73(–82) × 11–31 μm vs. 36–51 × 13.4–24 μm in *P. podospileus*) and smaller caulocystidia (36–125 × 6–25 µm vs. 36–140(–176) × 6–25 µm). There are 10 evolutionary changes between the nrITS sequences of *P. cystidiosus* and *P. podospileus* and 18 evolutionary changes in their *TEF1-α* sequences.

Based on current data, *P. podospileus* is not expected to occur in North America, and *P. cystidiosus* is not expected to occur in Europe, but it is not known if their geographical ranges overlap in Asia. *Pluteus cystidiosus* occurs in Japan and the Russian far east, while *P. podospileus* has been confirmed to occur as far east as Siberia (Krasnoyarsk Territory).

*Pluteus psychiophorus* (Berk. and Broome) Sacc. and *Pluteus stigmatophorus* (Berk. and Broome) Sacc. are two tropical species, originally described from Sri Lanka, for which molecular data are not currently available [18]. *Pluteus psychiophorus* differs from *P. cystidiosus* in the smaller basidiomata (pileus 7–20 mm), a smooth, non-squamulose stipe, larger basidiospores 6–8 × 5–6.5 μm, avl × avw = 6.5 × 5.7 μm, pleurocystidia predominantly clavate and smaller in size (35–48 × 18–23 μm), a heterogeneous lamellar edge not completely covered in cheilocystidia, and smaller elongated elements in the pileipellis, only up to 70 μm long [18]. *Pluteus stigmatophorus* differs in the pigmented lamellar edges and larger basidiospores 5.2–7.5 × 4.5–6 μm, avl × avw = 6.0 × 5.7 μm that are predominantly subglobose (Q = 1.05), predominantly clavate, pigmented, and shorter (up to 48 μm long) pleurocystidia, and pigmented, predominantly clavate, and shorter (up to 47 μm long) cheilocystidia [18]. *Pluteus delicatulus* C.K. Pradeep and K.B. Vrinda, originally described from India, differs in the smaller basidiomes (pileus 7–10 mm), smooth stipe without squamules, predominantly globose basidiospores, predominantly clavate and shorter (up to 40.5 μm) pleurocystidia, and the absence of caulocystidia [57].



***Pluteus inexpectatus*** Lebeuf, Justo and Ševčíková **sp. nov.** Figure 20 and Figure 21.

MB 849342.

*Etymology*: Named for the unexpected discovery of this unknown species in the *Pluteus podospileus* complex.

*Diagnosis*: Closely related to *P. seticeps*, differing in the presence of pleurocystidia, caulocystidia occurring in clusters, and different nrITS (14 evolutionary events) and *TEF1-α* (11 evolutionary events) sequences.

*Holotype*: *Canada*, Québec: Grondines, route Lefebvre, close to the railway track, on wood chips of angiosperm in a clear-cut area, 24 October 2021, leg. R. Lebeuf and A. Paul, HRL3639 (NBM-F-009793).

Pileus 10–21 mm in diameter, convex when young, expanding to applanate often with a low, broad umbo; surface uniformly granulose, sometimes rugose in the center, for the most part not cracked, rarely with a few cracks revealing the white context towards the margin or at mid-radius; with reddish brown to reddish blond colors (RAL color chart: beige (RAL 1211), mahogany brown (RAL 8016), darker at center; dry, not hygrophanous; margin entire, slightly sulcate; Lamellae close, free, convex, 2–3 mm broad, white when young, later pink, with even, serrate white edges. Stipe 12–21 × 1–2 mm, cylindrical, equal; surface white, almost smooth in the upper half and whitish-squamulose in the lower half, or minutely squamulose with dark or blackish brown squamules, more densely so in the lower half. Context in stipe and pileus greyish to concolorous with the outer surface, very thin in pileus. Smell and taste none.

Basidiospores [69/2/2] (4.2–)4.5–5.8(–6.0) × (4.0–)4.5–5.5(–5.8) μm, avl × avw = 5.0–5.1 × 4.7–5.0 μm, Q = 1.00–1.11, Q* = 1.03–1.07, globose to subglobose. Basidia 12–22 × 5–8 μm, 4-spored, clavate. Pleurocystidia mostly present close to the lamellar edge and then large, subglobose with or without pedicel, ovoid, ellipsoid; rare to absent elsewhere and then much smaller, clavate, ~23 × 12 µm; hyaline, thin-walled. Lamellar edge sterile. Cheilocystidia 18–45 × 8–24(–35) µm, mostly ellipsoid, (broadly) clavate or spheropedunculate, also ovoid or broadly fusiform; hyaline, thin-walled, crowded, forming a well-defined strip. Pileipellis a hymeniderm made up of two types of elements: (i) spheropedunculate, subglobose, ovoid-pedicellate, ellipsoid-pedicellate, 22–75 × 22–47 µm; (ii) (narrowly) fusiform, fusiform-pedicellate, lanceolate, broadly fusiform, narrowly utriform-pedicellate, conical, 67–165 × 13–39 µm; all elements with intracellular brown pigment aggregated in spots, thin-walled. Stipitipellis a cutis of cylindrical, thin-walled, 3–10 µm wide hyphae; hyaline. Caulocystidia of two types: (i) at stipe apex, scarce, isolated, or in small groups, 26–40(–80) × 11–16(–36) µm, ellipsoid, clavate, fusiform, sometimes catenulate, hyaline, also as brown-pigmented cylindrical terminal cells 2–3 µm wide with oblong or clavate extremity 62–190 × 11–19 µm; (ii) in lower stipe occasional or common, erect, often in clusters, more rarely isolated, 48–150 × 9–19 µm, (narrowly) fusiform, oblong, cylindrical, narrowly clavate, with rounded or attenuate apices, with intracellular brown pigment aggregated in spots. Clamp connections absent in all studied tissues.

Habit, habitat, and phenology: isolated or gregarious, growing on well-decayed wood or wood chips of angiosperms. Thus far, it has been collected in forests or open areas of the temperate zone in September and October.

Distribution: North America: molecularly confirmed from Canada (Québec).

Additional collections examined: **CANADA**. **Québec:** Saint-Narcisse, Parc de la rivière Batiscan, Secteur Barrage, along the road leading to trail Le Buis, predominantly deciduous forest of *Quercus rubra*, *Pinus strobus*, *Acer* sp., and *Abies balsamea*, on well-decayed wood of angiosperm, 8 September 2022, leg. R. Lebeuf and A. Paul, HRL4010 (NBM-F-009794).

Notes: *Pluteus podospileus* differs from *P. inexpectatus* in its subglobose to broadly ellipsoid basidiospores and distribution restricted to Eurasia; some collections of *P. podospileus* have larger basidiomata, but the variability in size of *P. inexpectatus* is not well known yet. *Pluteus cystidiosus* can have a more robust habit, a more or less velvety pileus, and subglobose to broadly ellipsoid basidiospores. In North America, confusion is most likely with *Pluteus seticeps*, which can be distinguished by the absence or very rare pleurocystidia even close to the lamellar edge and caulocystidia mostly not occurring in clusters. Another North American species, *P. absconditus*, differs by its pulverulent-powdery and/or white-translucent pubescent pileus, larger basidiospores, heterogeneous lamellar edge, and growth on soil.



***Pluteus millsii*** Justo, Borovička, Grootmyers, Kalichman and S.D. Russell **sp. nov.**
Figure 22 and Figure 23.

MB 849343.

*Etymology*: At the request of the original collector, Jan Borovička, this species is dedicated to Michael “Mike” Edward Mills, basist, pianist, and background vocalist of the band R.E.M. Mike is considered R.E.M.’s little genius, and we describe this little American *Pluteus* in his honor.

*Diagnosis*: Closely related to *Pluteus necopinatus*, differing in the presence of pleurocystidia, shorter cheilocystidia, occurrence in the Northern Hemisphere, and different nrITS sequences (17 evolutionary events).

**Holotype: USA. Connecticut:** New Haven, Yale University Campus, Morse College Garden, on very humid, decayed plant matter, 15 July 2012, leg. Jan Borovička, AJ 838 (NBM-F-009803). The holotype was found at the venue of the Annual Meeting of the Mycological Society of America.

Pileus 5–10 mm in diameter, hemispherical when young, expanding to convex or plano-convex, with a low, broad umbo; surface strongly granulose-squamulose all over, with cracks revealing the white context underneath, except at the center; with predominant brown colors (Munsell: 5YR 4/6, 3/4; 7.5YR 5/6–5/8, 4/6), sometimes gray-brown (7.5YR 6/3–6/6), darker at center; dry, not hygrophanous; margin entire, rimose, or very slightly translucently striate, Lamellae crowded, free, ventricose, up to 2 mm broad, white when young, later pink, with white flocculose edges. Stipe 10–20 × 1–2 mm, cylindrical, with a slightly broadened subbulbous base (up to 3 mm wide); surface white, silvery-white, shiny, and entirely hairy-woolly, covered with white hairs, sometimes grouped in definite mats with pale brown tips. Context in stipe and pileus white. Smell and taste none.

Basidiospores [90/4/3] (4.5–)4.9–5.8(–6.2) × (4–)4.4–5.8 μm, avl × avw = 5.2–5.6 × 4.4–5.1 μm, Q = 1.00–1.26, Q* = 1.10–1.18, globose to broadly ellipsoid. Basidia 16–24 × 6–9 μm, 4-spored, clavate. Pleurocystidia 36–47 × 17–22 μm, mostly clavate or narrowly clavate, some ovoid or ellipsoid; hyaline, thin-walled; scarce and scattered all over lamellar faces. Lamellar edge sterile. Cheilocystidia 22–39 × 10–21 μm, mostly (narrowly) clavate or (narrowly) utriform, more rarely ovoid, in some collections a few rostrate, with an apical projection up to 14 μm long; hyaline, thin-walled, crowded, forming a well-developed strip. Pileipellis a hymeniderm or trichohymeniderm, made up of two types of elements: (i) narrowly clavate, narrowly utriform, or fusiform, 30–85 × 17–25 µm; (ii) fusiform, narrowly utriform, cylindrical, 90–170 × 10–22 µm, often with 1–3 septa at the base; all elements with brown intracellular pigment, slightly thick-walled. Stipitipellis a cutis of cylindrical, slightly thick-walled, 7–10 µm wide hyphae, hyaline, or rarely some with pale brown pigment. Caulocystidia absent, but above the external layer of the stipitipellis often with loosely arranged tufts of hair-like, parallel to ascending, hyphal elements, with terminal elements 65–107 × 12–22 µm, cylindrical, narrowly clavate, narrowly utriform, mostly hyaline, a few with pale brown pigment, some with oily, slightly refractive content; thin-walled to slightly thick-walled. Clamp connections absent in all studied tissues.

Habit, habitat, and phenology: subgregarious, growing on well-decomposed wood and other plant matter. June–September.

Distribution: Eastern North America: so far, only known from the USA (Connecticut, Kentucky, and Tennessee).

Additional collections examined: **USA. Kentucky:** Bell Co., Clear Creek Springs, in a dry section of a creek bed, 20 September 2019, leg. Jack Steven Smith, iNaturalist 34927657. **Tennessee:** Knox Co., Knoxville, Knoxville Botanical Garden and Arboretum, 22 September 2018, leg. Jacob Kalichman, iNaturalist 112187791 (NBM-F-009804); ibid., Third Creek Greenway Park, on a well-rotted, decorticated hardwood log with *Trechispora regularis*, *Thanatephorus* sp., and *Oliveonia pauxilla*, within a stand of *Phyllostachys aurea*, 29 June 2021, leg. Django Grootmyers, Mushroom Observer 458003 (NBM-F-009805).

Notes: *Pluteus millsii* stands out macroscopically by the densely hairy-woolly surface of the stipe. *Pluteus absconditus* can also have a relatively hairy stipe, but this species differs in the presence of 2-spored and 1-spored basidia, slightly larger pleurocystidia (up to 66 µm long), a heterogeneous lamellar edge, and common caulocystidia.

Molecularly, *P. millsii* is quite distinct from all other taxa in the/podospileus clade: its nrITS sequences differ by 19 evolutionary changes from those of its closest relative, *Pluteus necopinatus* (96.9% sequence similarity). All other *Pluteus* nrITS sequences sampled in this study are less than 90% similar to *P. millsii*.

*Pluteus necopinatus* differs from *P. millsii* by the non-hairy-woolly stipe surface, the absence of pleurocystidia, the larger cheilocystidia (37–74 × 12.5–30 μm) that are predominantly lageniform or utriform, and a pileipellis with shorter elongated elements (up to 125 μm long). Both taxa have similar tufts of hyphal elements above the stipitipellis, which we do not consider to be caulocystidia in the sense the term is applied in *Pluteus*, as they are loosely arranged bundles of hyphae that lay on top of the tightly arranged hyphae of the stipitipellis. In other taxa of *Pluteus* with true caulocystidia, these arise directly from the stipitipellis. *Pluteus necopinatus* was originally described and is thus far only known from the Atlantic Forest of Brazil, growing directly out of a concrete wall [46].



***Pluteus gausapatus*** Ševčíková and Antonín **sp. nov.** Figure 24 and Figure 25.

MB 849804.

Etymology: *gausapatus* refers to the plush pileus and stipe surface. “*Gausapatus”* also means “happy”, which well expresses the authors feelings about enjoying this beautiful species.

*Diagnosis*: *Pluteus gausapatus* notably differs from *Pluteus podospilloides* by the absence of the following features: khaki, dark brown, or olive-brown squamules and white short hairs on a pileus surface; dark brown squamules of the stipe; utriform or broadly lageniform colored pleurocystidia with a short neck; and by a distinct phylogenetic position (nrITS, *TEF1-α*) among the *Pluteus* species in the/podospileus clade.

Holotype: Republic of Korea: Taean Peninsula, Deoksung, Sudeoksa Temple, mixed forest, on decaying wood of broadleaved tree, 197 a.s.l., 8 July 2014, leg. V. Antonín (BRNM 817745).

Pileus 9–10 mm broad, convex with slightly depressed center and inflexed margin, not hygrophanous, smooth, entirely pruinose-tomentose to “plush”, hairy at the margin, indistinctly striate only up to 1 mm from the margin, nut brown (RAL 8011), slightly paler towards the margin. Lamellae free, L = 32, l = 1–3, distant, ventricose, up to 3 mm broad, whitish with a pale pinkish tinge, with an irregularly nut-brown flocculose edge. Stipe 10–12 × 1–1.5 mm, cylindrical, not bulbous, curved, entirely finely pubescent to “plush”, finely longitudinally fibrillose, white, in the lower part minutely squamulose with indistinct, pale brown floccules. Context whitish. Smell and taste indistinct.

Basidiopores [60/1/2] (4.8–)5.0–6.0(–6.5) × (4.0–)4.2–5.5 μm, avl × avw = 5.4–5.7 × 4.7–5.2 µm, Q = 1.00–1.25(–1.33), Q* = 1.13; globose to broadly ellipsoid, thick-walled, inamyloid. Basidia 19–26 × 7–10(–11) μm, clavate to subfusiform, 4-spored, very rarely 2-spored. Pleurocystidia (43–)51–62(–68) × (20–)23–27(–31) μm, scattered, mostly fusiform, thin-walled, hyaline. Lamellar edge sterile. Cheilocystidia 25–76 × 12–26 μm, variable in shape: vesiculose, ovoid, clavate to broadly clavate, subfusiform to broadly fusiform, mostly with a short neck, thin-walled, with brown intracellular pigment, rarely hyaline. Pileipellis composed of abundant elongate elements 100–170 × (10–)11–17(–20) μm, cylindrical with subacute apex or subfusiform, mixed with shorter broadly fusiform elements with obtuse or subacute apex 40–90 × 22–28 μm and short clavate to ovoid elements 20–45 × 9–26 μm; all elements thin-walled with brown intracellular pigment. Stipitipellis a cutis of whitish hyphae 4–9(–11) μm wide. Caulocystidia abundant, in tufts, 40–60(–90) × 12–18 μm, clavate to fusiform, hyaline, in the lower part also with pale brownish intracellular pigment. Clamp connections absent in all studied tissues.

Habit, habitat, and phenology: solitary, on a decayed thick branch of a broadleaved tree, July.

Distribution: Known only from Asia: Republic of Korea.

Notes: *Pluteus gausapatus* is characterized by its small basidiomata with a plush, brown pileus, brown lamellar edges, a whitish, finely pubescent to plush stipe, mostly fusiform pleurocystidia, and growth on decayed deciduous trees.

*Pluteus podospilloides* E.F. Malysheva and O.V. Morozova, originally described from Vietnam, differ by their pileus surface covered with minute khaki, dark brown, or olive-brown squamules and white short hairs; a stipe with a distinct dark brown squamule along its entire length; utriform or broadly lageniform colored pleurocystidia with a short neck; slightly thick-walled cheilocystidia; and pileipellis elements [28]. *Pluteus delicatulus* C.K. Pradeep and K.B. Vrinda is somewhat similar to *P. gausapatus* by its small basidiomata and pileipellis, but it differs by its lamellae without brown edges, glabrous stipe without caulocystidia, slightly broader basidiospores, and vesiculose to clavate pleurocystidia [57]. *Pluteus stigmatophorus* (Berk. and Broome) Sacc. has similar pileipellis elements to *P. gausapatus*, *but its pileus is larger and depressed at the center, its stipe is brownish yellow and black-dotted, its basidiospores are slightly larger, and its pleurocystidia are clavate-pedicellate to submucronate* [8,16,18]. *Pluteus stigmatophorus* is known from Sri Lanka [17]. *Pluteus extremiorientalis* E.F. Malysheva and Malysheva also have a brownish-colored and tomentose-squamulose pileus, brownish to fuscous lamellar edges, and a stipe (lower part) covered by dark brown squamules (or fibrils). However, this Russian Far East species differs by having a fibrillose-squamulose pileus margin, larger basidiospores, predominantly lageniform or utriform pleurocystidia, and a pileipellis consisting of only two types of shorter elements [58].



***Pluteus notabilis*** Eyssart. and Ševčíková **sp. nov.**, Figure 26 and Figure 27.

MB 849805.

Etymology: from the Latin adjective *notabilis*, “noteworthy, remarkable”.

Diagnosis: *Pluteus notabilis* differs from *Pluteus inflatus* by its pileipellis with fusoid to utriform elements with a long mucronate apex, some brownish flocculose lamellar edges, smaller basidiospores on average, and a sterile lamellar edge.

Holotype: France. Doubs. Les Fourgs, on soil with woody debris and sawdust of co-nifers (*Abies* and/or *Picea*), a heap formed to store snow piled up during the winter, according to the “snowfarming” technique, with *Psathyrella scanica*, *Psathyrella suavissima,* and *Cystoderma superbum* among other species, 25 September 2018, leg. G. Eyssartier, holotype PC0142588, isotype BRNM 825869.

Pileus 18–45 mm broad, convex or hemispherical when young, later plano-convex; surface completely and remarkably felted, often fairly coarsely radially wrinkled, not striate or only shortly translucently striate on overmature specimens, with fimbriate or denticulate margin, not or very weakly hygrophanous, warm date brown to almost blackish brown when young (RAL 8022), paler and with chestnut brown to reddish brown tinges on adult specimens, dark brown with distinctly paler margin on very old specimens. Lamellae moderately distant, free, ventricose, white when young, then pinkish, with an eroded-denticulate edge, with a whitish, less frequently brownish flocculose edge, especially near the pileus margin. Stipe 17–48 × 2–5 mm, cylindrical, with a slightly enlarged base, whitish to cream, with a very pale, indistinct ochre tinge on very old specimens, densely covered with distinct brown and very fine floccules, often aligned in longitudinal stripes. Context white to whitish; smell and taste indistinct; basal mycelium white. Spore print pink.

Basidiospores (4.5–)4.7–5.4(–5.8) × (3.4–)3.6–4.5(–5.0) μm, avl × avw = 5 × 4 μm, Q = 1.1–1.5, Q* = 1.26, broadly ellipsoid to ellipsoid or subglobose, rarely globose. Basidia (18–)20–28(–32) × 6–9(–11) μm, 4-spored, narrowly clavate or narrowly fusiform, hyaline. Pleurocystidia (29–)43–80(–90) × (17–)17–25(–35) μm, scattered, moderately abundant, broadly fusiform to utriform or clavate, rarely almost spheropedunculate or subovoid. Lamellar edge sterile. Cheilocystidia (18–)25–90(–105) × (10–)11–25(–30) μm, subcylindrical to narrowly clavate or clavate, less frequently subutriform, hyaline, or with brown intracellular pigment. Pileipellis a hymeniderm composed of three types of elements, all colored brown by intracellular pigment: (i) clavate to spheropedunculate elements measuring (28–)30–50 × 20–40 µm, composing a fairly regular sublayer; (ii) fusiform to (sub-)utriform elements, 50–82 × 15–27 µm, with a more or less conical or distinctly mucronate apex, mucron up to 8 μm long, some mucrons constricted; frequently with thickened wall on upper part; (iii) long subcylindrical emerging elements up to 180(–250) × 15 µm, thin-walled on exsiccata, giving a fluffy appearance to areas of the pileus where they are abundant. Stipitipellis composed of cylindrical and thin-walled hyaline hyphae (3–)5–10(–15) μm wide. Caulocystidia numerous, in tufts, (40–)45–145 × 15–30(–35) μm, thin-walled, predominantly long, subcylindrical, and reminding the long elements of the pileipellis, but also clavate or (sub-)utriform. Clamp connections absent in all studied tissues.

Habit, habitat, and phenology: on soil with woody debris and sawdust of conifers and on strongly decomposed wood near fallen trunks of *Abies alba*. August–September.

Distribution: Europe: France and the Czech Republic.

Additional collections examined: **CZECH REPUBLIC**: Beskydy Mountains, Razula Nature Reserve, *Abieto*-*Fagetum* with *Picea abies*, on strongly decomposed wood near a fallen trunk of *Abies alba*, near also *P. granulatus*, 15 August 2019, leg. H. Ševčíková, BRNM 825870.



***Pluteus notabilis* var. *insignis*** Ševčíková, Cáfal, Ferisin and Galbusera **var. nov.** Figure 28 and Figure 29.

MB 849806.

Etymology: from the Latin adjective *insignis*, “remarkable, noteworthy”.

Diagnosis: Differs from *Pluteus notabilis* var. *notabilis* by having lamellae without brown edges, larger basidiospores on average, shorter pleurocystidia, and pileipellis elements without a distinct mucronate apex.

Holotype: SLOVAKIA: Chvojnická pahorkatina, Vieska-Horáreň, on decaying deciduous wood, 28 August 2016, leg. R. Cáfal (BRNM 825868).

Pileus 20–35 mm broad, convex to campanulate when young, later plano-convex to almost plane, with obtuse center without or with low umbo, dull; surface finely granulose to granulose-velvety, some finely tomentose in the center, not or indistinctly wrinkled or sparsely veined radially up to the edge, without striation, with slightly inflexed margin, not hygrophanous, warm date brown, reddish brown, dark brown to almost warm blackish brown when young (RAL 8017 or darker, RAL 8022), then slightly paler. Lamellae L = 36–48, l = 1–5, free, crowded to moderately distant to almost distant, ventricose, white when young, then pinkish to flesh pink, with an eroded-denticulate, concolorous, finely flocculose edge. Stipe 38 × 1.5–2.5 mm, cylindrical, slightly broadened toward the lower part, with a slightly enlarged base, solid, innately fibrillose, twisted or not, hyaline, whitish, or cream, graying with age, entirely minutely to densely covered with distinct brown fine floccules, sometimes aligned in short longitudinal stripes, at least on the lower part. Context whitish in pileus, whitish or hyaline in stipe. Smell and taste indistinct. Spore print pinkish.

Basidiospores [93/3/3] (4.0–)4.5–6.6(–7.0) × 3.5–5.6(–6.5) μm, avl × avw = 5.65 × 4.4 μm, Q = 1.0–1.5, Q* = 1.21–1.32, subglobose to broadly ellipsoid, rarely globose or ellipsoid, thick-walled. Basidia (17–)18–29 × 6–10 μm, 4-spored, narrowly clavate to narrowly fusiform, hyaline, thin-walled. Pleurocystidia 26–60(–62) × 14–30(–32) μm, scattered, rare, broadly clavate to clavate, subovoid to ovale-oblong or narrowly to broadly fusiform to utriform or subutriform, often pedunculate, hyaline, thin-walled. Lamellar edge sterile. Cheilocystidia (17–)20–82(–93) × (10–)11–30(–34) μm, numerous, subcylindrical to narrowly clavate to broadly clavate or vesiculose, subfusiform or fusiform with short neck and obtuse apex, hyaline, thin-walled. Pileipellis a hymeniderm composed of short and elongate elements: (i) (28–)30–55(–82) × 20–40(–45) µm elements broadly clavate to clavate or spheropedunculate, rarely subfusiform to fusiform, some submucronate; (ii) narrowly (sub)fusiform or narrowly cylindrical elements 90–175 × (16–)18–23(–26) µm, some with long neck up to 35 µm; all elements with brown intracellular pigment, with thin or slightly thickened wall. Stipitipellis a cutis composed of cylindrical hyphae (2.5–)5–10(–14) μm wide, hyaline, thin-walled, or with slightly thickened hyphae. Caulocystidia (54–)75–170 × 6.5–15(–28) µm, numerous, in tufts, mostly narrowly to broadly fusiform, more rarely clavate, cylindrical or subcylindrical, with brown intracellular pigment, thin-walled. Clamp connections absent in all examined tissues.

Habit, habitat, and phenology: on soil in riparian forests and on decaying deciduous wood, including *Populus tremula*. May–August.

Distribution: Europe: Italy, Slovakia, and Spain.

Additional collections examined: **ITALY**: Matelica, Marche, on stump of *Populus tremula*, 28 October 2022, leg. Sofie and A. Galbusera (AG 22210280). **SPAIN**: Gavarres mountains, Fitor (city), in riparian forest with *Corylus avellana*, *Alnus glutinosa,* and *Quercus ilex*, on siliceous soil, about 400 m a.s.l., 18 May 2006, leg. C. Roqué (originally det. as *P. podospileus*, BRNM825867).

Notes: fusiform to utriform pileipellis elements with a mucronate apex up to 8 μm are confirmed only for *Pluteus notabilis* var. *notabilis* and for *Pluteus seticeps*. The later pleurocystidia are absent or extremely rare, and this species occurs in North America. *Pluteus notabilis* var. *insignis* differs from *Pluteus notabilis* var. *notabilis* by its concolorous lamellar edges, somewhat larger basidiospores on average, pleurocystidia only up to 58 µm long, and pileipellis without elements displaying a distinct, long mucronate apex. Moreover, *Pluteus notabilis* var. *notabilis* has a distinctly radially wrinkled pileus, while some basidiomata lack this feature and some are less distinctly wrinkled. However, the variability of this feature needs to be confirmed. Phylogenetically, based on nrITS and *TEF1-*α, both taxa are similar.

Macroscopically, *Pluteus notabilis* somewhat resembled the original description of *Pluteus granulatus* Bres. by its granulose brownish pileus and its growth on *Abies alba* [59]. However, our holotype study confirms that this species belongs to section *Hispidoderma*. The similar *Pluteus stigmatophorus* (Berk. and Broome) Sacc. also has fusoid to utriform pileipellis elements, sometimes resembling the pileocystidia of *Pluteus notabilis* var. *notabilis*, but they are never mucronate [18]. Moreover, *Pluteus stigmatophorus* has broader, subglobose to globose basidiospores; cheilo- and pleurocystidia with a dark pigment; a pileus depressed at the center; a black-dotted stipe on a brownish yellow ground [8,17,18], and it is known from tropical Sri Lanka [17]. *Pluteus anomocystidiatus* Menolli and de Meijer [60], known from Brazil, has short (29–33 × 16.2–20 µm) clavate or spheropedunculate pleurocystidia, long (50–116 × 12.5–25 µm) narrowly fusiform or filiform cheilocystidia with distinct brownish pigment macroscopically visible as a distinct dark brown lamellar edge, and a different pileipellis.




**Excluded, insufficiently known and doubtful taxa**


***Pluteus praestabilis*** (Britzelm.) Sacc.

Sylloge Fungorum 5: 672, 1883.

Basionym: *Agaricus praestabilis* Britzelm., Berichte des Naturhistorischen Vereins Augsburg 27: 193, 1883.

This species was originally described as having a blackish brown velvety pruinose pileus 45 mm in diameter, a white stipe measuring 45 × 5 mm with brown floccules, basidiospores measuring 6 × 4–5 µm and as growing on soil near *Fagus sylvatica* [8,61]. Stangl and Bresinsky [62] considered *Pluteus praestabilis* a synomym of *P. plautus*. Although this taxon may belong to the *Pluteus plautus* complex, it may also represent a species related to the *P. podospileus* complex [53]. All the characteristics mentioned resemble several species belonging to the *Pluteus podospileus* group. Unfortunately, the protologue is vague, the holotype does not exist, and, therefore, its identity cannot be satisfactorily clarified.

## 4. Discussion

In this study, we show the *P. podospileus* complex to be considerably more diverse than previously believed. Although some overlap in the morphological variation of the species does occur, the distribution patterns and macroscopic aspects often include characters that allow identification at the species rank. Macroscopically, *P. gausapatus* differs from all similar species by the combination of a plush pileus surface and a dark lamellar edge. *P. notabilis* can also have some darker lamellar edges, but its pileus is granulose-velvety rather than plush with a wrinkled center. Furthermore, the whole stipe is densely covered with distinct, fine brown floccules, often aligned in longitudinal stripes. *P. millsii* is remarkable for its densely hairy-woolly stipe surface and strongly granulose-squamulose pileus all over, with cracks that reveal the white context underneath. *P. podospileus* was originally described with a rugulose pileus center. We keep the original concept of *P. podospileus* and use this name for a clade in which a strongly rugulose pileus center is a typical feature. However, *P. notabilis*, some collections of *P. inflatus* and *P. cutefractus*, and the North American *P. seticeps*, *P. cystidiosus,* and *P. absconditus* can also have a rugulose pileus, at least when young. Typical basidiomata of *P. fuscodiscus* have a distinct darker disc in the pileus center. However, several species commonly possess a darker pileus center. Thus, microscopical examination is needed to identify *P. fuscodiscus*.

*Pluteus cutefractus* is characterized by a markedly cracked pileus, but as the pileus of several species of the *P. podospileus* group may be cracked, a combination of features must be used to identify this species. The pileus margin of *P. seticeps* is often translucently striate up to half the radius. However, *P. cystidiosus* may also share this feature. The most macroscopically variable species is *P. inflatus*. *P. absconditus* is often macroscopically indistinguishable from *P. inflatus*, but a pileus with a pulverulent-powdery aspect and/or a white-translucent pubescence can be useful if present.

Spores of *P. inflatus*, *P. absconditus,* and *P. cutefractus* are commonly slightly longer (up to ± 7 μm), while basidiospores of *P. inexpectatus*, *P. cystidiosus*, *P. millsii,* and *P. notabilis* do not exceed 6 μm. However, this is a relatively subtle difference. Basidiospores of *P. inexpectatus* seem to often be almost globose to subglobose (Q* = 1.03–1.07), but more collections are needed to confirm the stability of this feature.

Pleurocystidia are an important feature for the delimitation of *Pluteus* species. Pleurocystidia are absent or extremely rare in *P. seticeps* and absent or present in *P. inflatus* and *P. fuscodiscus*. They are present (from scattered to common) in the other species described here.

The lamellar edge is sterile in several species, while a heterogeneous and partially fertile lamellar edge is typical for *P. inflatus* and *P. absconditus* and also occurs in some collections of *P. cutefractus*. Cheilocystidia are numerous in all species of the *P. podospileus* complex and variable across almost all species. Cheilocystidia tend to be utriform to spathulate in *P. fuscodiscus*, while they are more often subglobose, ovoid, or ellipsoid in *P. inexpectatus* and mostly clavate or narrowly clavate in *P. millsii*, but this unstable feature is not sufficient for a definitive identification of individual species.

Further investigation will confirm whether the colorful cheilocystidia of *P. gausapatus* can be a stable character similar to, for example, *P. umbrosus* (Pers.) P. Kumm. [4,63] or, on the contrary, if it is variable and unreliable as in *P. keselakii* Ševčíková, P.-A. Moreau, and Borovička [31].

In most species, the pileipellis is composed of two types of elements (short and elongate). Favre [54] described *P. granulatus* var. *tenellus* as having a pileipellis composed of three types of elements: (i) short, broadly clavate to spheropedunculate and (sub)fusiform, (ii) elongate cylindrical, and (iii) long, elongate (sub)lanceolate to subfusiform or subcylindrical. However, there are several options on how to classify these elements as being of two or three types. In fact, in some collections, there is a fine line between whether we can describe two or three types of elements. Fusiform to (sub-)utriform elements with a distinctly mucronate apex up to 8 μm long in the pileipellis are a unique feature of *P. notabilis* var. *notabilis*.

Caulocystidia are present in most species but might be lacking in some particular collections. *Pluteus millsii* and *P. necopinatus* have a layer of loosely arranged hyphae on top of the stipitipellis that we do not consider caulocystidia here, as they are not terminal elements that rise from the stipitipellis. Some authors have considered the presence/absence of caulocystidia a reliable feature, distinguishing *Pluteus nanellus* from *P. seticeps* [2]. *P. inflatus* frequently has long caulocystidia up to 145 μm, while the caulocystidia of *P. cutefractus* and *P. gausapatus* are shorter. Our studies indicate that this feature could be used as an additional character to help define species on morphological grounds, but it might have limited usefulness to identify individual collections due to its inconstant presence within collections of the same species.

Based on phylogenetically confirmed collections, the distribution area of *P. podospileus* may be presumed to be limited to Eurasia, or at least Europe and Siberia. In Europe, six species are phylogenetically confirmed: *Pluteus podospileus*, *P. inflatus*, *P. cutefractus*, *P. fuscodiscus*, *P. notabilis,* including *P. notabilis* var. *insignis*, and an undescribed species represented by only one collection (GM 3751) in our tree. The range of at least three species, *P. podospileus*, *P. inflatus*, and *P. fuscodiscus*, extends to northwestern Asia. Because both *P. cystidiosus* and *P. fuscodiscus* were also found in the Russian Far East, we expect these species to have wider distributions than currently documented. *P. gausapatus* is only known from the Republic of Korea, where an additional, probably undescribed species, *P.* aff. *podospilloides*, was also collected. Eight North American species are recognized in our tree (Figure 1). Only *P. seticeps* and *P. cystidiosus* were previously described. *P. millsii*, *P. absconditus*, and P. *inexpectatus* are described in this article as new, while more collections will be necessary to describe the remaining species.

The present paper represents a big step forward in our understanding of the taxonomy of *P. podospileus* and related taxa, but it also shows that more work is needed to fully describe the species diversity of this group. Four single-collection taxa sampled during this study are included in the phylogenies (Figure 1). Molecularly, they seem to represent separate species from the ones described here, but we await additional collections to formally describe them. It should be noted that these yet-undescribed species come from areas that are heavily sampled in this study, like Europe (*Pluteus* sp. GM3751) and Eastern North America (*Pluteus* sp. *MO270622; Pluteus* sp. *iNaturalist 13936381; Pluteus* sp. *iNaturalist 27406926*).

Most of the species described here appear on decayed wood (stumps or trunks) of angiosperms or on woody debris. *P. inflatus*, *P. cutefractus,* and *P. notabilis* var. *insignis* can have a terrestrial growth habit, but they can also grow on decayed wood. In the protologue of *P. granulatus* var. *tenellus*, which represents *P. inflatus*, an association with *Filipendula ulmaria* is mentioned. *P. absconditus* grows on soil, often among leaves, twigs, and other plant matter, but without connection to wood. However, more collections are needed to verify its habitat. Only *P. fuscodiscus* was found on both coniferous and deciduous wood. Moreover, it was also found on litter in fern thickets. *P. notabilis* was only found on well-decayed coniferous wood or woodchips.

## Identification Key

Species occurring in North America*

1. Pleurocystidia absent……………………………………………………….……………… 21. Pleurocystidia present (scattered to common)…………………………………………… 32. Average spore length 5.3–6.2 μm. In ruderal, man-made habitats (e.g., urban gardens). Known from Quebec (Canada) and Washington (USA).…………………………*P. inflatus*2. Average spore length 4.6–5.0 μm. Mostly in forest, non-ruderal, habitats. Widely distributed in Eastern North America…………………………………….…………… *P. seticeps*3. Stipe distinctly hairy, hairy-woolly, with hairs often grouped in mats with pale brown tips. Stipitipellis with loosely arranged tufts of hair-like, parallel to ascending, hyphal elements, with terminal elements 65–107 × 12–22 µm………………….………… *P. millsii*3. Stipe pubescent to hairy, or covered with distinct brown floccules. Stipitipellis with clusters of caulocystidia rising from the stipitipellis………………………………….…… 44. Basidia 4-, 2- and 1-spored. Lamellar edge heterogeneous, partially fertile with a mix of basidia, basidioles and cheilocystidia…………………………………….… *P. absconditus*4. Basidia exclusively 4-spored. Lamellar edge sterile, exclusively with cheilocystidia. 55. Pileus 14–35(–40) mm. Basidiospores average Q = 1.11–1.24. Pleurocystidia 35–73(–82) × 11–31 μm, mostly (narrowly) utriform……………………………………… *P. cystidiosus*5. Pileus 10–21 mm. Basidiospores average Q = 1.03–1.07. Pleurocystidia 23–50 × 14–34 μm, mostly subglobose (with or without pedicel) or clavate……………… *P. inexpectatus*



Species occurring in Eurasia*

1. Pileus plush, stipe white, lamellar edge dark………………………………. *P. gausapatus*1. Combination of features different…………………………………………………….……. 22. Pileus granulose velvety with a wrinkled centre, lamellar edge dark, stipe with brown floccules, pileipellis with some mucrons up to 8 μm long………. *P. notabilis* var. *notabilis*2. Combination of features different…………………………………………………….……. 33. Pileus rugulose, basidiospores small, up to 6.0(–6.4) × 5.1(–5.2) μm, pleurocystidia up to 52 × 26 μm …………………………………………………………………….…*P. podospileus*3. Pileus not rugulose or basidiospores or/and pleurocystidia larger……………………..44. Lamellar edge heterogeneous, partially fertile ………………………………………….54. Lamellar edge sterile…………………………………………………………………………65. Pileus cracked, caulocystidia up to 145 μm long………………………….*P. cutefractus*5. Pileus not or only slightly cracked, caulocystidia up only to 60 μm long..…*P. inflatus*6. Pileus centre with a distinct darker disc, pleurocystidia absent or short, up to 36 μm, mostly utriform to spathulate………………………………………………… *P. fuscodiscus*6. Pileus centre without a distinct darker disc and/or pleurocystidia longer, often with different shape…………………………………………………………………………………77. Pileus cracked, caulocystidia short, up to 60 μm ….…………………………*P. cutefractus*7. Pileus not or indistinctly cracked, caulocystidia longer……………………………………………88. Basidiospores small, up to 5.5(–6.2) × 5.0 μm, Russian Far East……………*P. cystidiosus*8. Basidiospores larger, up to 6.5(–7.0) μm, Europe……………….… *P. notabilis* var. *insignis*

*Only the described species are included in the keys

## Figures and Tables

**Figure 1 jof-09-00898-f001:**
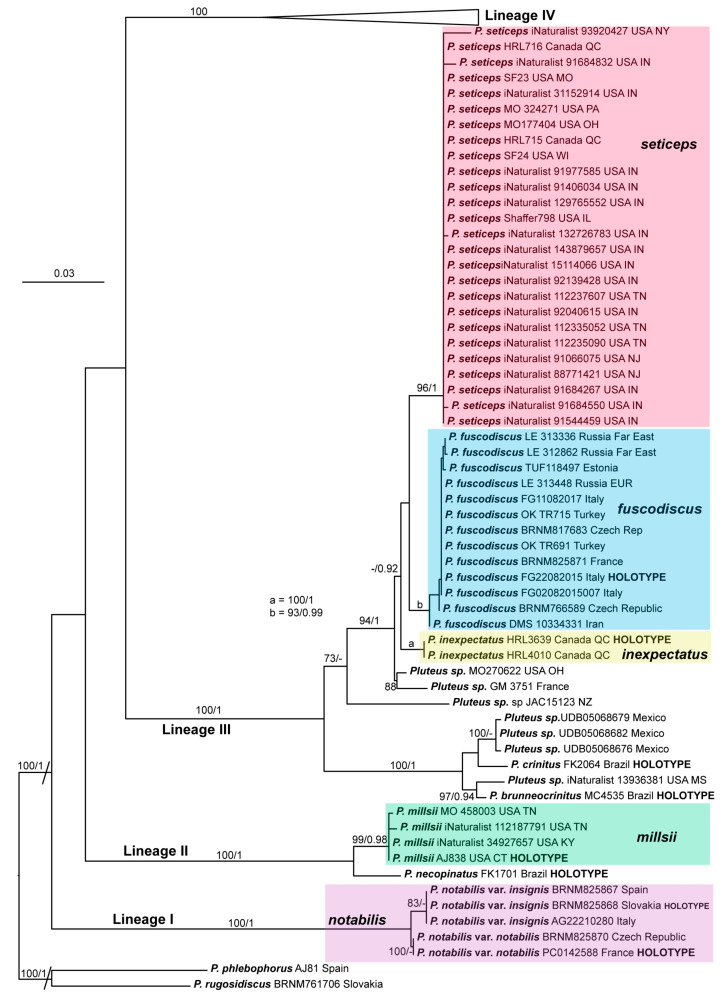
Best tree from the ML analysis of the nrITS + *TEF1*-α dataset of the/podospileus clade. BS ≥ 70 and PP ≥ 0.90 are shown on or below the branches. The root length has been reduced to facilitate graphical representation.

**Figure 2 jof-09-00898-f002:**
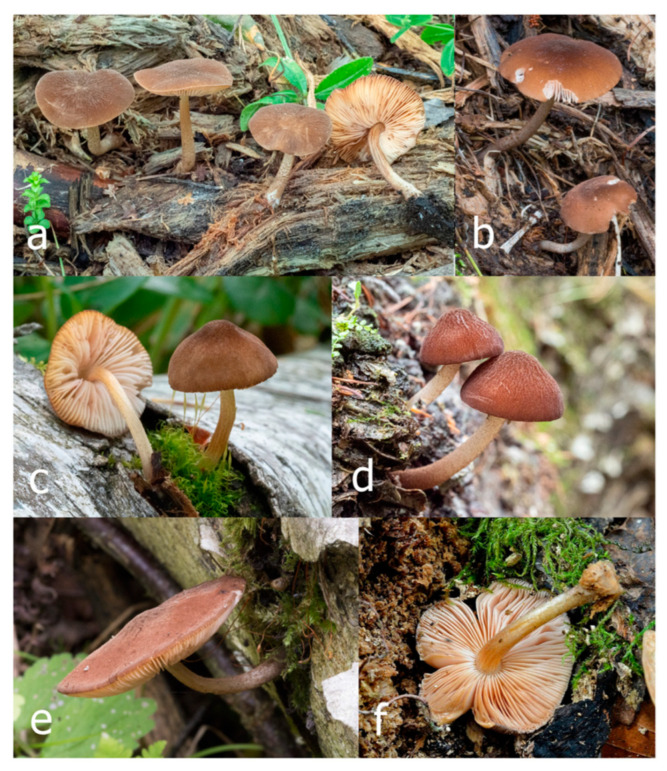
*Pluteus podospileus* basidiomata in situ: (**a**) LE F-313589, (**b**) LE F-303682, (**c**) LE F-313611, (**d**) LE F-313615, (**e**) LE F-303687, (**f**) BRNM825840. Photo: (**a**–**e**) E. F. Malysheva, (**f**) P. Včelička.

**Figure 3 jof-09-00898-f003:**
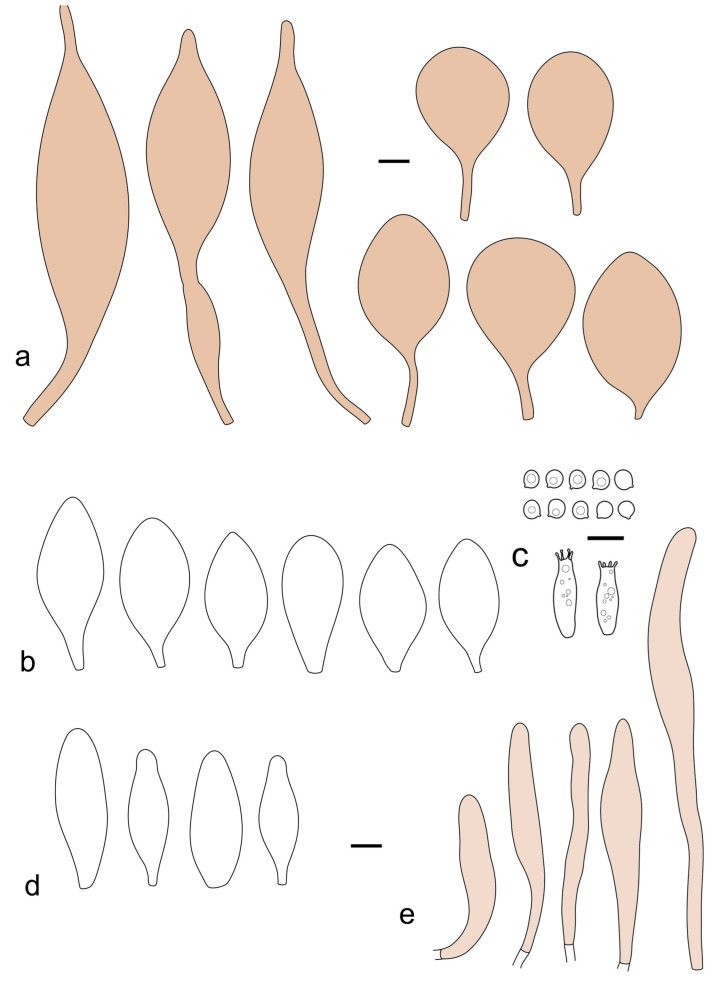
*Pluteus podospileus* LE F-303682 (**a**) pileipellis elements, (**b**) cheilocystidia, (**c**) basidia and basidiospores, (**d**) pleurocystidia, (**e**) caulocystidia. Scale bar 10 μm.

**Figure 4 jof-09-00898-f004:**
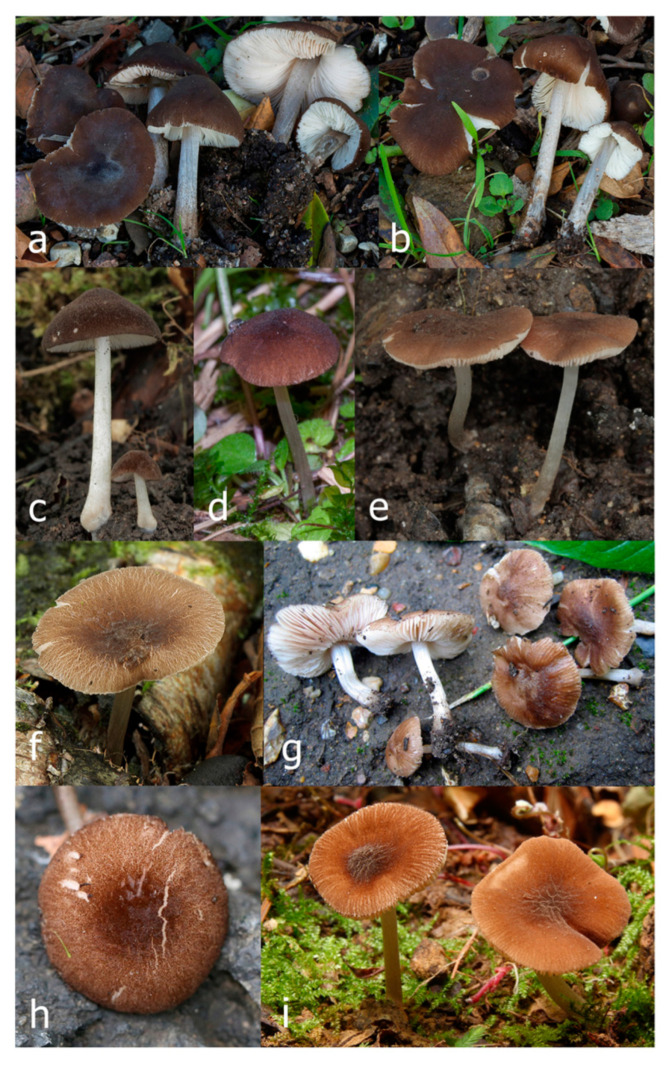
*Pluteus inflatus* basidiomata in situ: (**a**,**b**) DMS 10041511, (**c**) BRNM817761, (**d**) LE F-262712, (**e**) BRNM825838, (**f**) DMS-10027661, (**g**) GM1524, (**h**) OK-TR-260, (**i**) FG250620179.

**Figure 5 jof-09-00898-f005:**
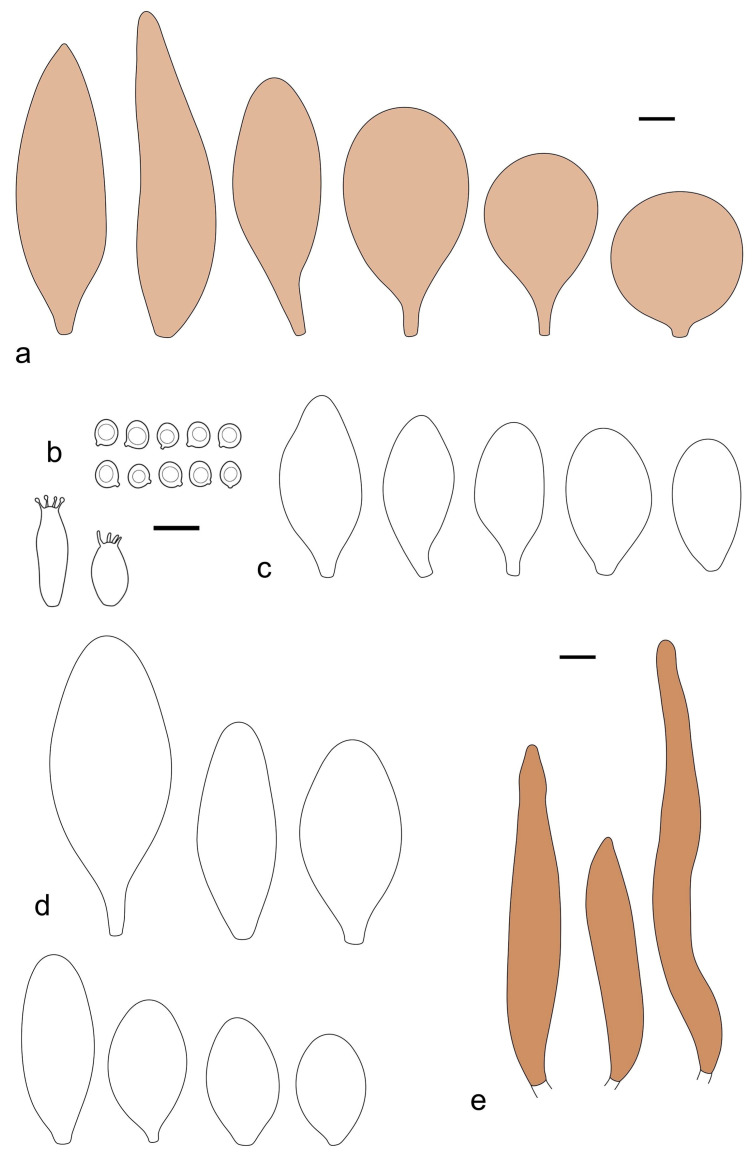
*Pluteus inflatus* LE F-262712. (**a**) pileipellis elements, (**b**) basidia and basidiospores, (**c**) pleurocystidia, (**d**) cheilocystidia, (**e**) caulocystidia. Scale bars 10 μm.

**Figure 6 jof-09-00898-f006:**
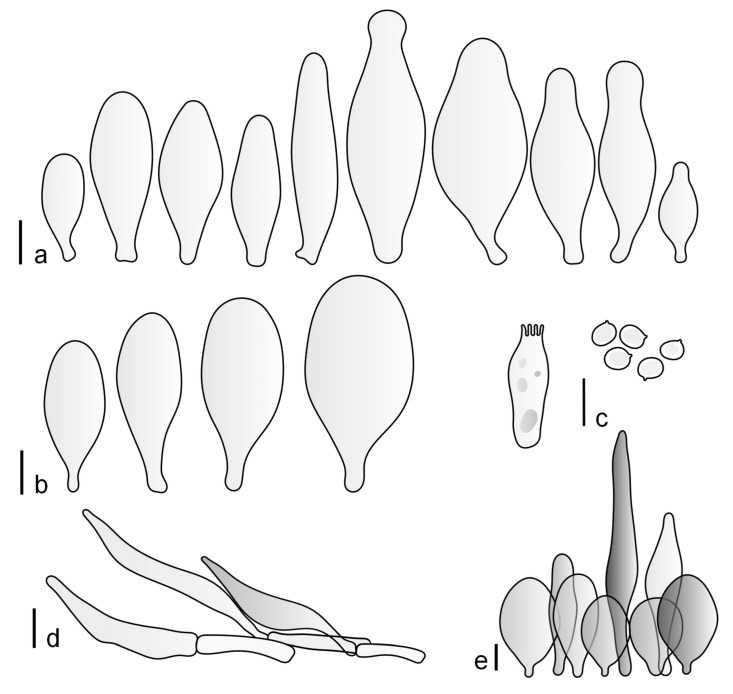
*Pluteus inflatus* FG250620179. (**a**) cheilocystidia (**b**) pleurocystidia, (**c**) basidia and basidiospores (**d**) caulocystidia, (**e**) pileipellis elements. Scale bars 10 μm.

**Figure 7 jof-09-00898-f007:**
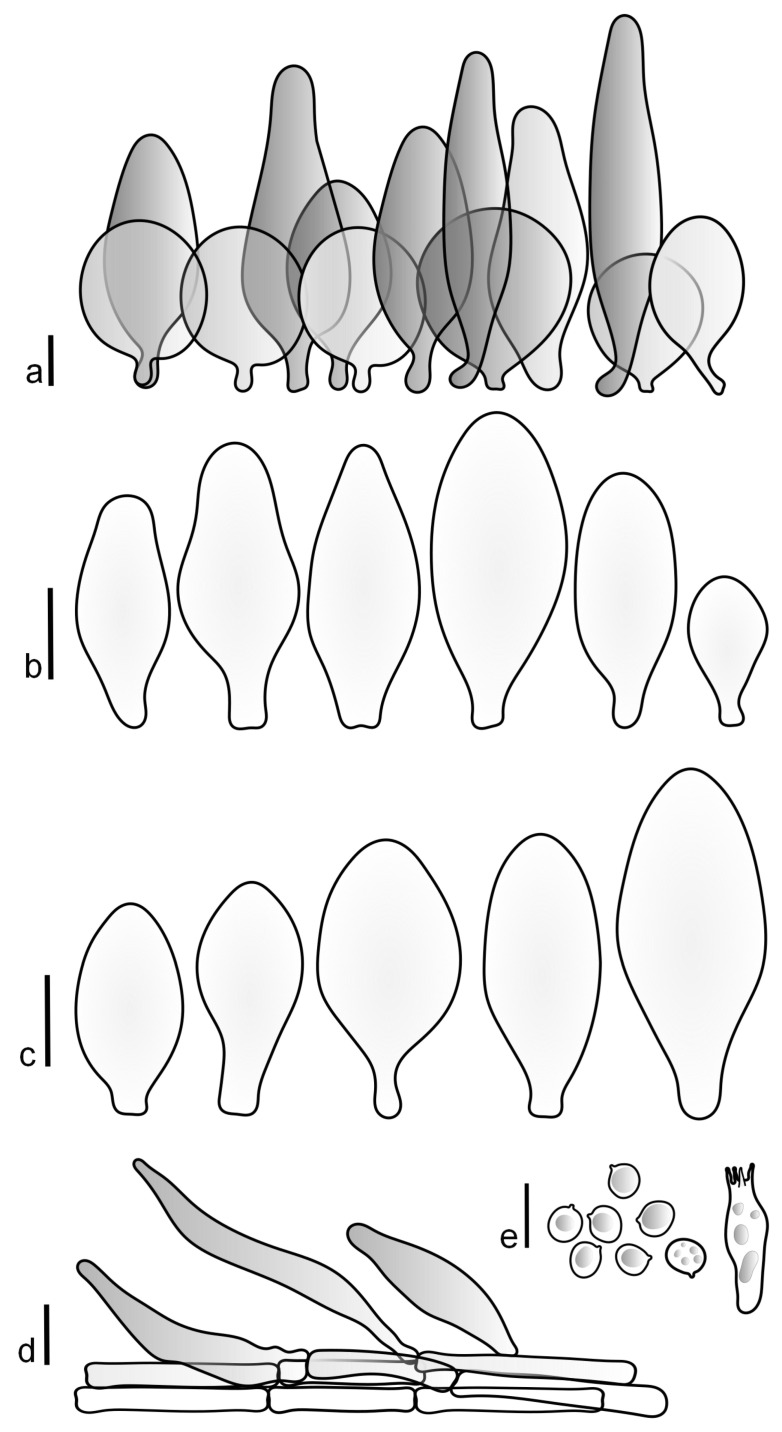
*Pluteus inflatus* FG 16092018026. (**a**) pileipellis, (**b**) cheilocystidia, (**c**) pleurocystidia, (**d**) caulocystidia, (**e**) basidia and basidiospores. Scale bars 10 μm.

**Figure 8 jof-09-00898-f008:**
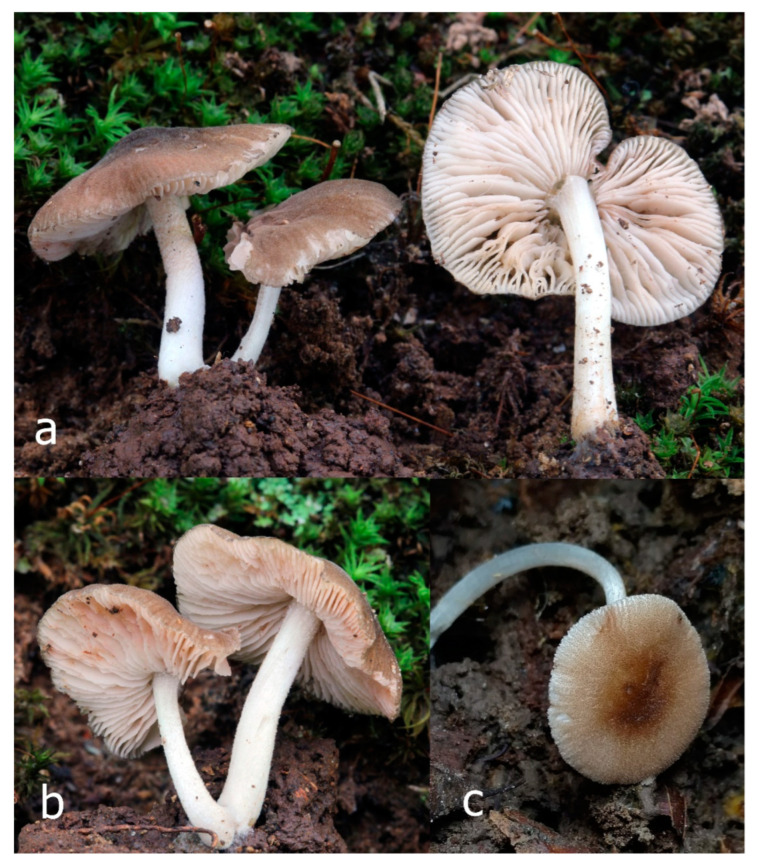
*Pluteus absconditus* basidiomata in situ: (**a**,**b**) MO 136488, (**c**) iNaturalist 112240775.

**Figure 9 jof-09-00898-f009:**
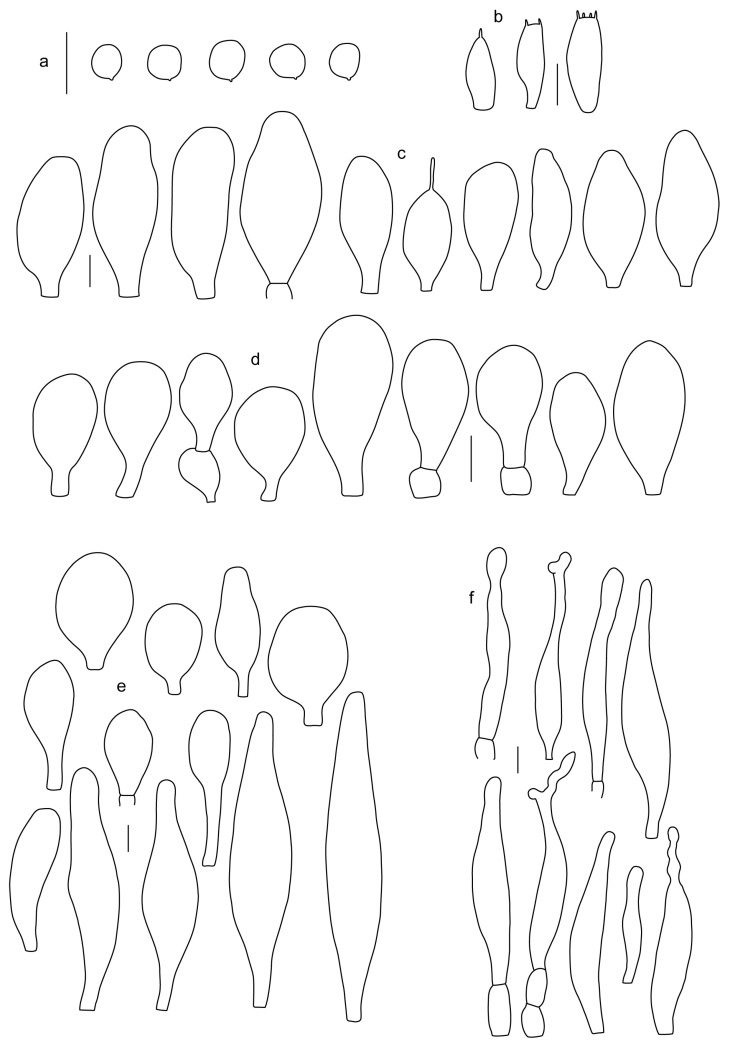
*Pluteus absconditus* MO136488 (**a**) basidiospores. (**b**) basidia. (**c**) pleurocystidia. (**d**) cheilocystidia. (**e**) pileipellis elements. (**f**) caulocystidia. Scale bars 10 μm.

**Figure 10 jof-09-00898-f010:**
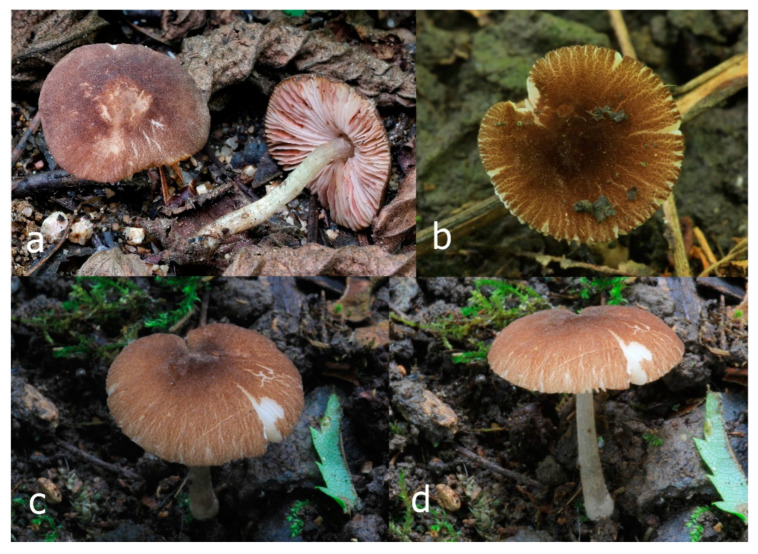
*Pluteus cutefractus* basidiomata in situ: (**a**) GM3458, (**b**–**d**). BRNM 816221.

**Figure 11 jof-09-00898-f011:**
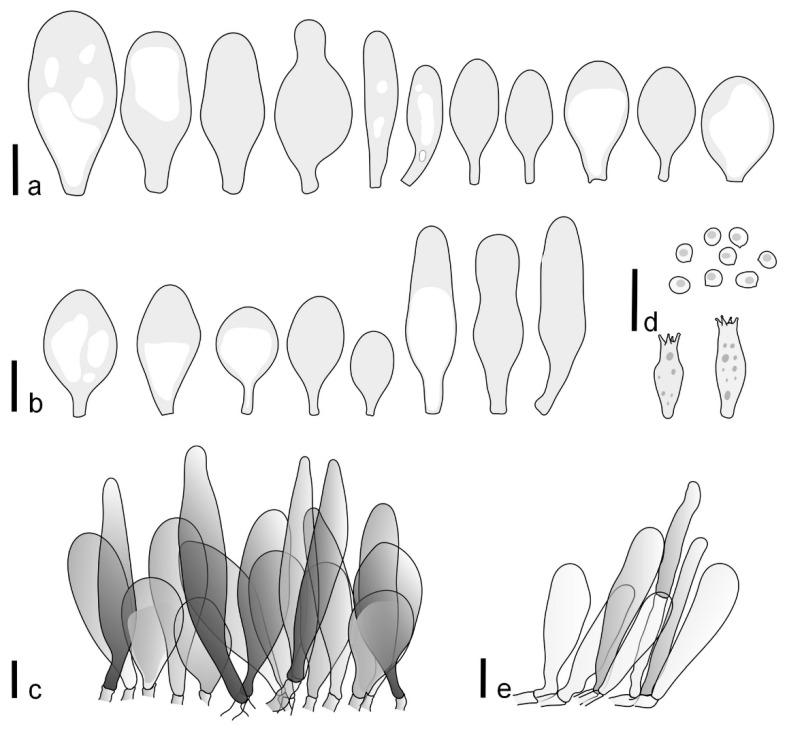
*Pluteus cutefractus* MCVE 30110 holotype. (**a**) cheilocystidia, (**b**) pleurocystidia, (**c**) pileipellis, (**d**) basidiospores and basidia, (**e**) stipitipellis. Scale bars 20 μm.

**Figure 12 jof-09-00898-f012:**
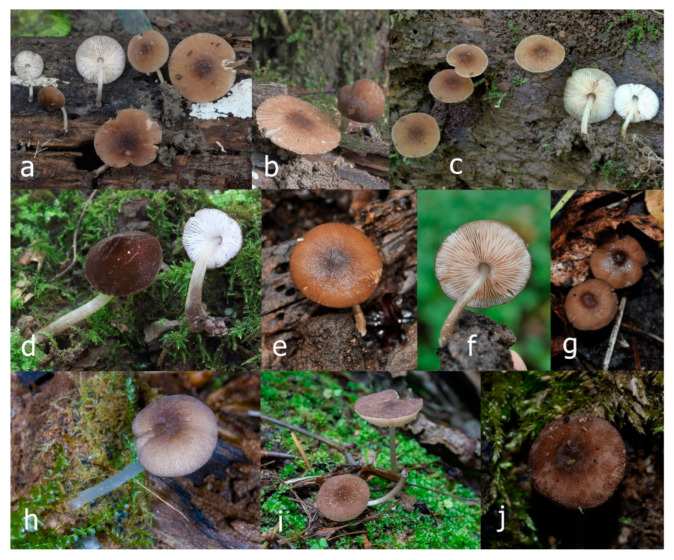
*Pluteus fuscodiscus* basidiomata in situ: (**a**,**b**) FG 653, (**c**) ALV 12152, (**d**) FG22082015. (**e**,**f**) OKA-TR715, (**g**) OKA-TR691, (**h**) LE F-313336, (**i**) LE F-312862, (**j**) BRNM825871.

**Figure 13 jof-09-00898-f013:**
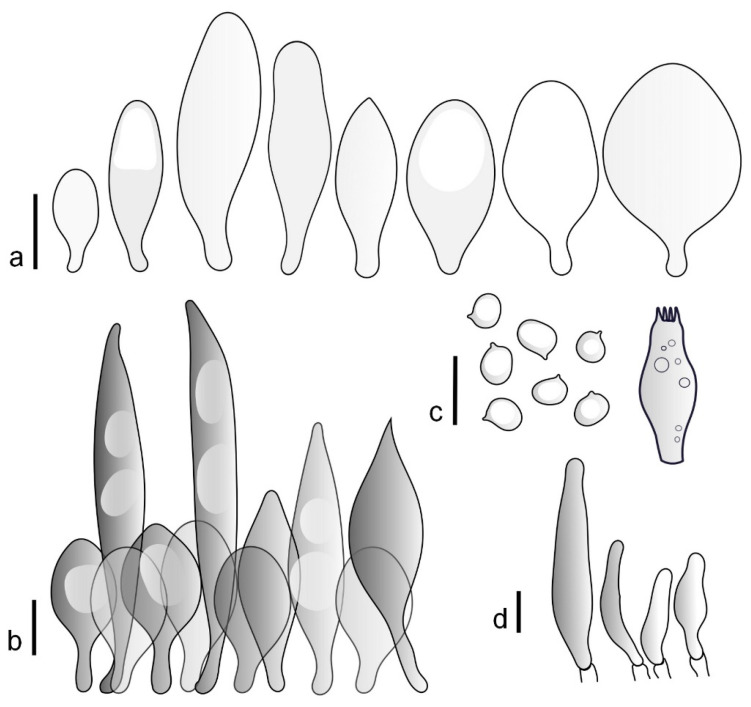
*Pluteus fuscodiscus* holotype. (**a**) cheilocystidia, (**b**) pileipellis elements, (**c**) basidiospores and basidium, (**d**) caulocystidia. Scale bars 10 μm.

**Figure 14 jof-09-00898-f014:**
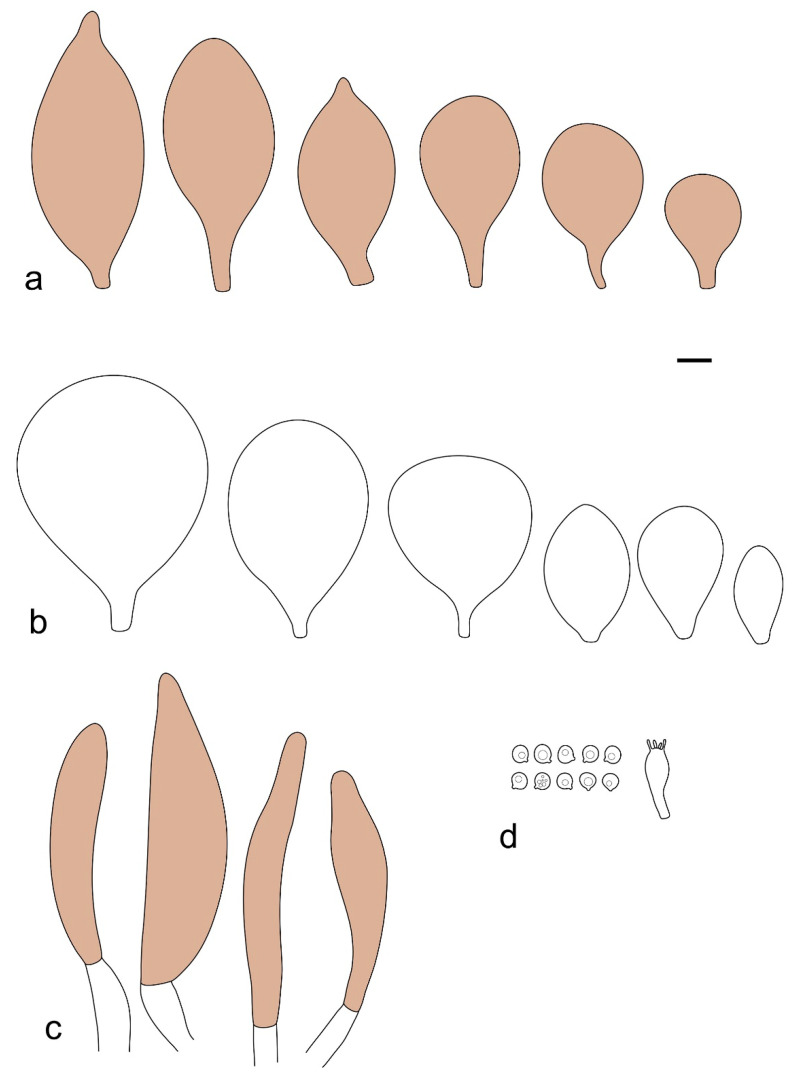
*Pluteus fuscodiscus* LE F-312862. (**a**) pileipellis elements, (**b**) cheilocystidia, (**c**) caulocystidia, (**d**) basidium and basidiospores. Scale bar 10 μm.

**Figure 15 jof-09-00898-f015:**
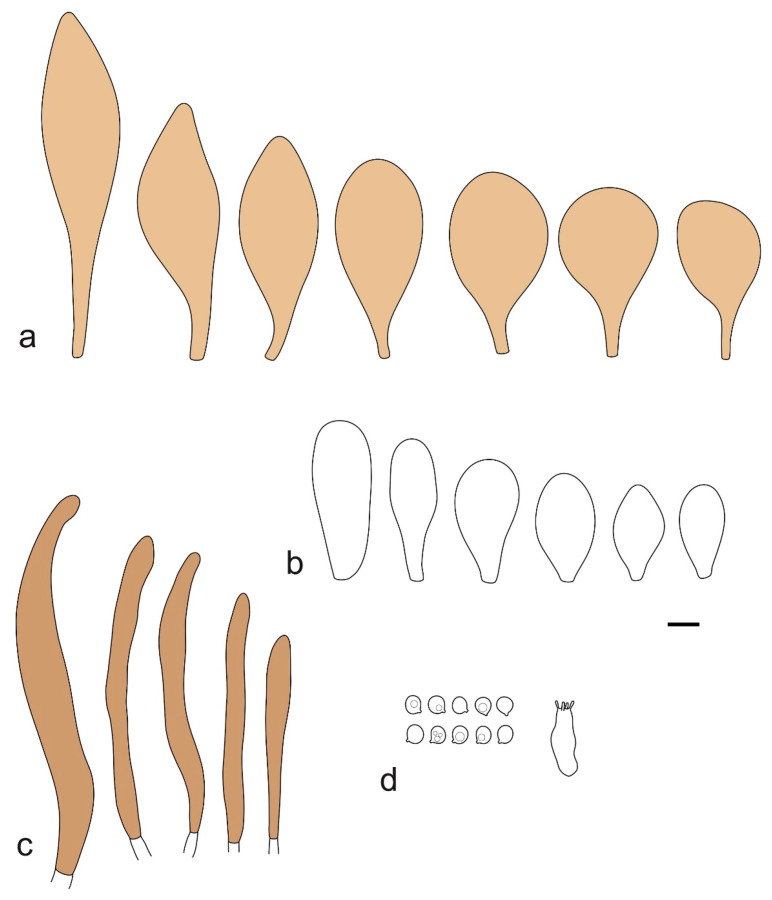
*Pluteus fuscodiscus* LE F-313448. (**a**) pileipellis elements, (**b**) cheilocystidia, (**c**) caulocystidia, (**d**) basidium and basidiospores. Scale bar 10 μm.

**Figure 16 jof-09-00898-f016:**
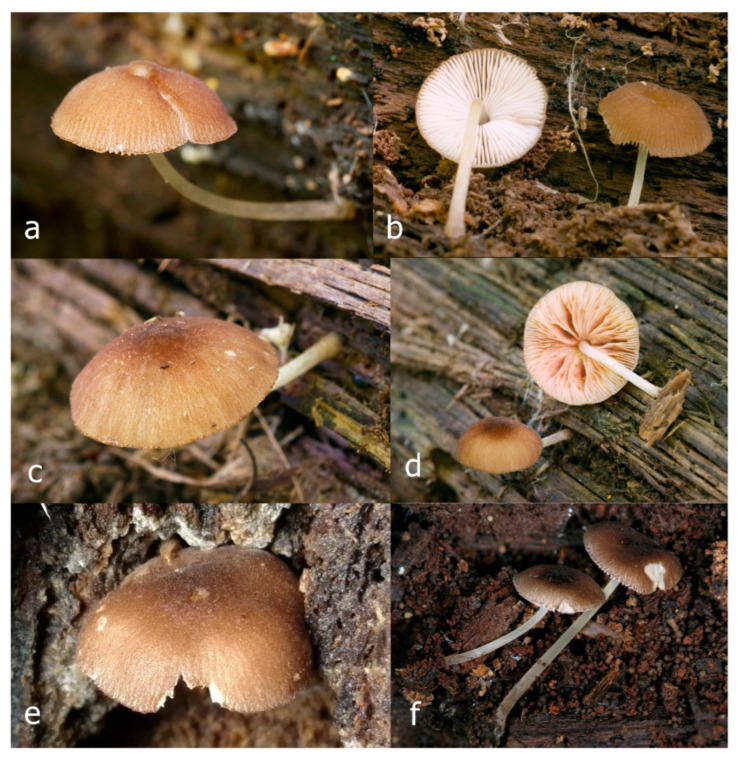
*Pluteus seticeps* basidiomata in situ: (**a**,**b**) HRL0715, (**c**,**d**) HRL0716, (**e**) iNaturalist 112237607, (**f**) iNaturalist 112335052.

**Figure 17 jof-09-00898-f017:**
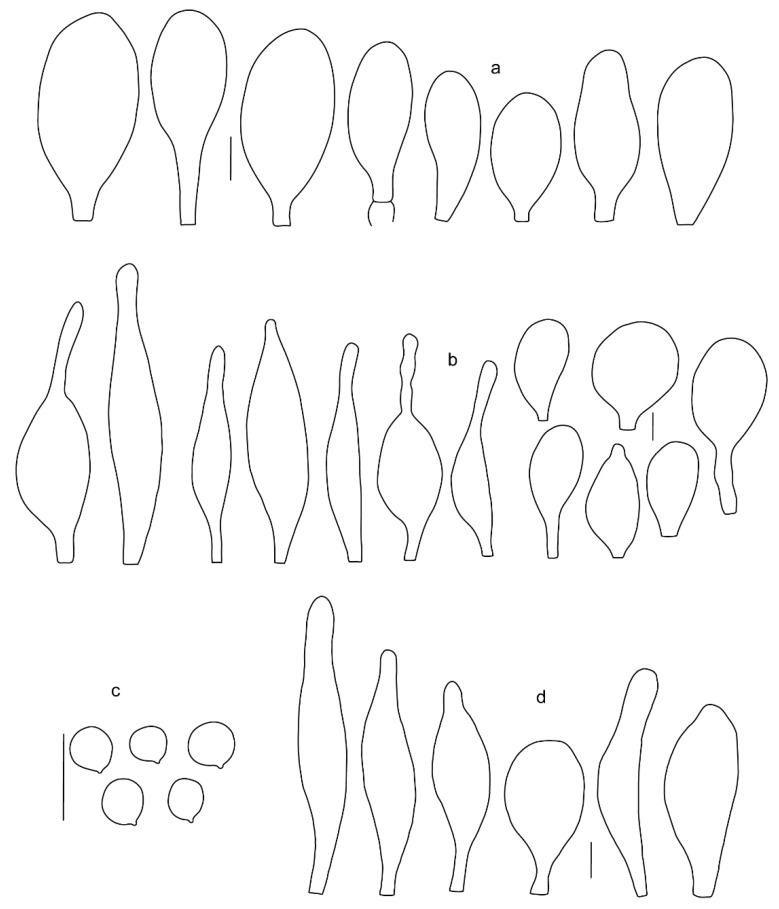
*Pluteus seticeps* MO177404. (**a**) Cheilocystidia. (**b**) pileipellis elements. (**c**) basidiospores. (**d**) caulocystidia. Scale bars 10 μm.

**Figure 18 jof-09-00898-f018:**
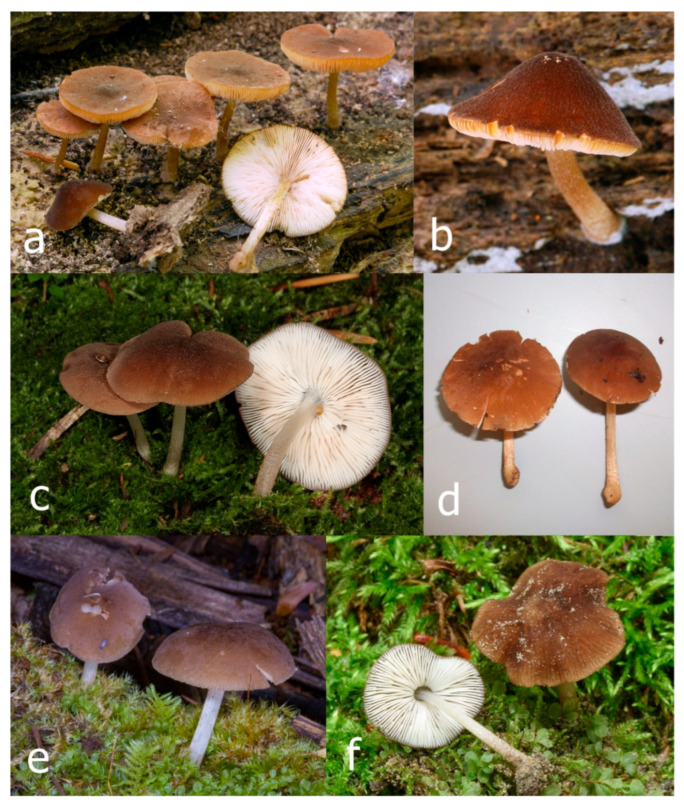
*Pluteus cystidiosus* basidiomata in situ: (**a**,**b**) NBM-F-009792, (**c**) LE F-313335, (**d**) NBM-F-009790, (**e**) LE F-312852, (**f**) NBM-F-009791.

**Figure 19 jof-09-00898-f019:**
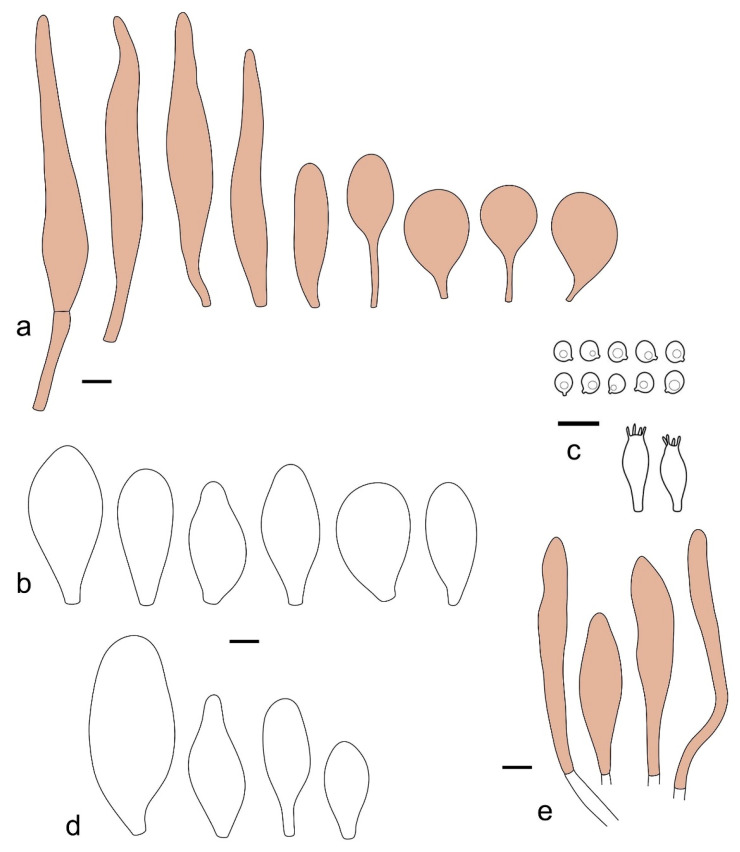
*Pluteus cystidiosus* LE F-313335. (**a**) pileipellis elements, (**b**) cheilocystidia, (**c**) basidia and basidiospores, (**d**) pleurocystidia, (**e**) caulocystidia. Scale bars 10 μm.

**Figure 20 jof-09-00898-f020:**
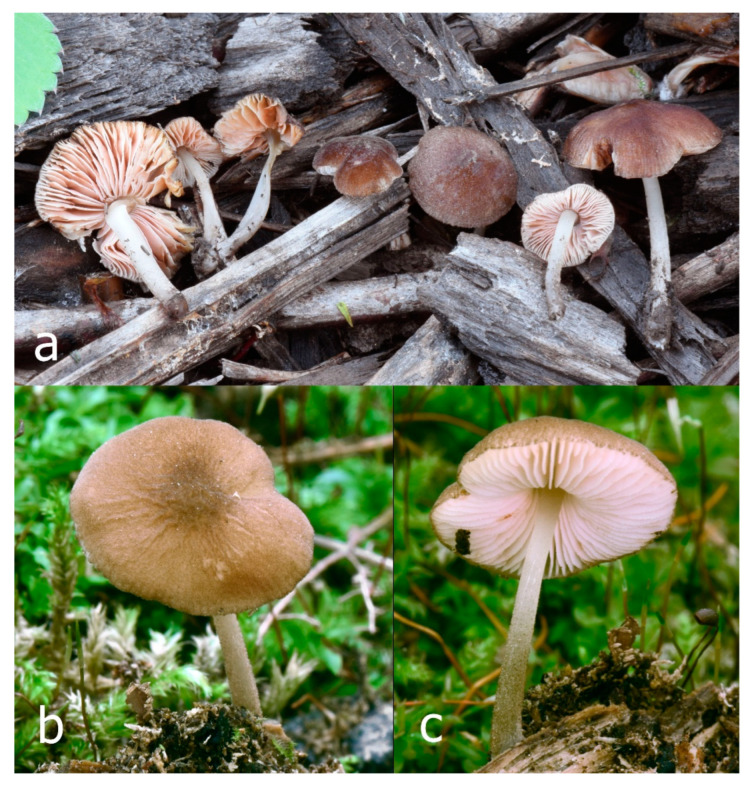
*Pluteus inexpectatus* basidiomata in situ: (**a**) NBM-F-009793, (**b**,**c**) NBM-F-009794.

**Figure 21 jof-09-00898-f021:**
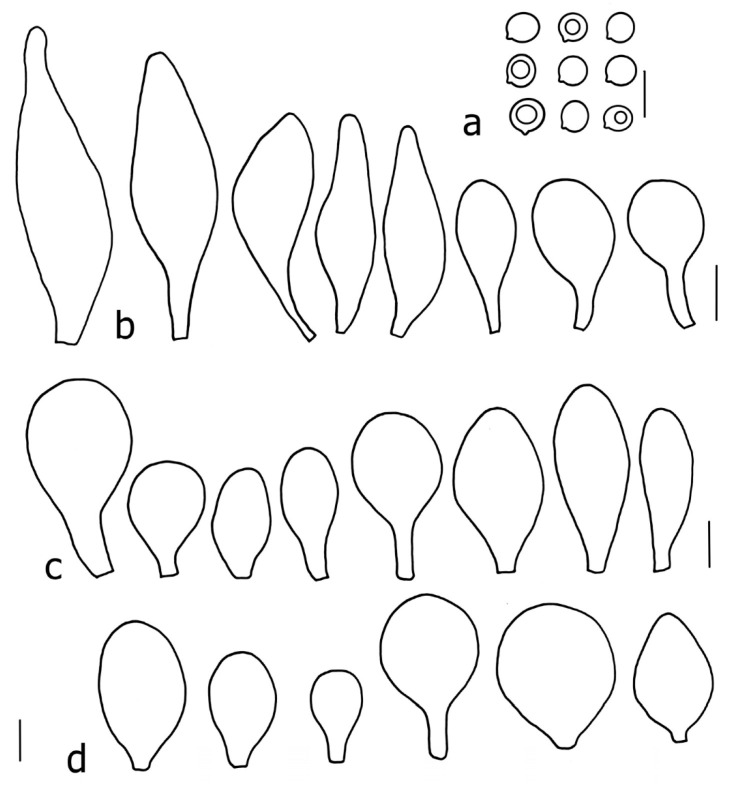
*Pluteus inexpectatus* holotype: (**a**) basidiospores, (**b**) pileipellis elements, (**c**) cheilocystidia, (**d**) pleurocystidia. Scale bars 10 μm.

**Figure 22 jof-09-00898-f022:**
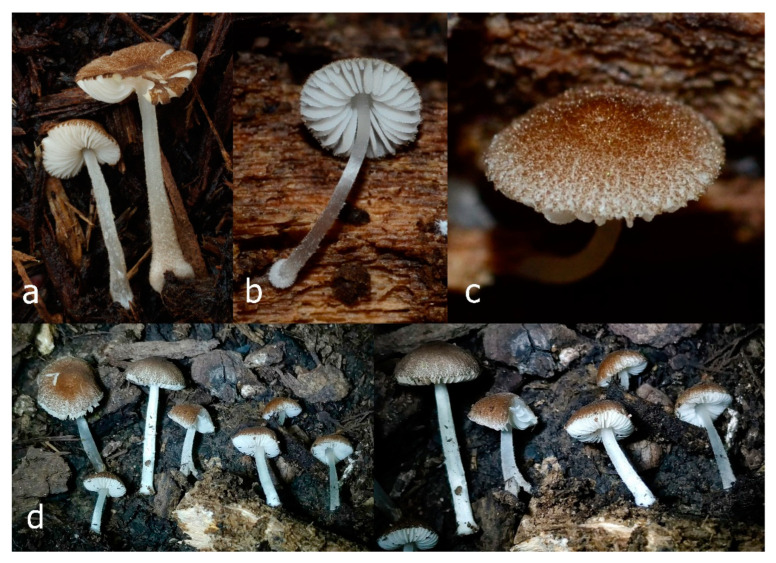
*Pluteus millsii* basidiomata in situ: (**a**). holotype NBM-F-009803, (**b**,**c**). MO 458003, (**d**). NBM-F-009804.

**Figure 23 jof-09-00898-f023:**
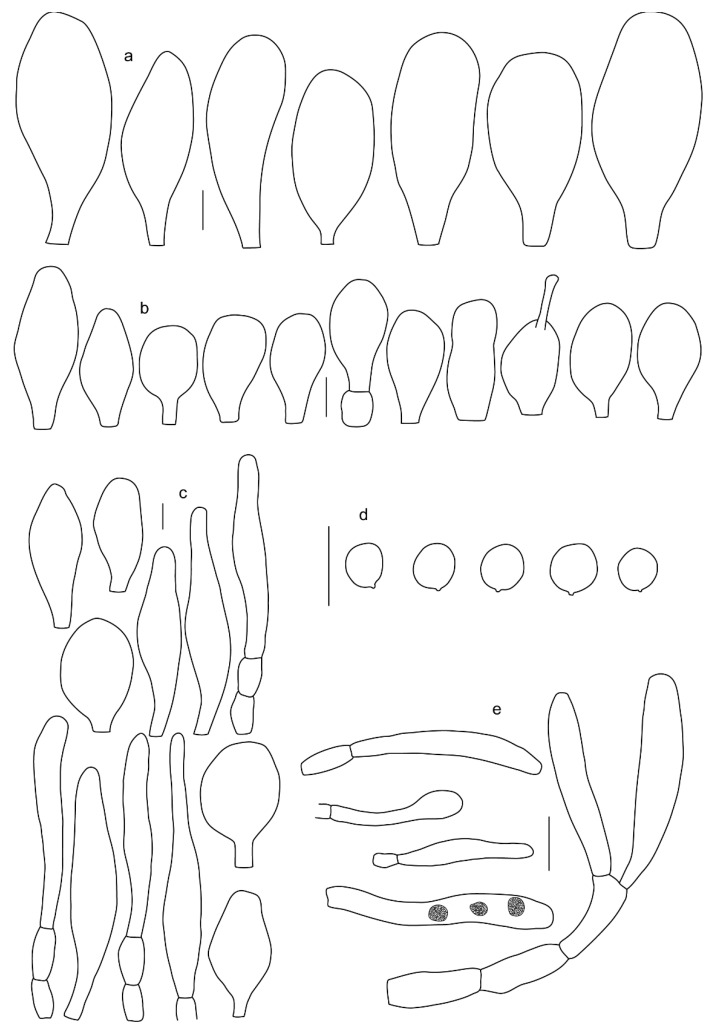
*Pluteus millsii* holotype. (**a**). Pleurocystidia. (**b**). Cheilocystidia. (**c**). Pileipellis elements. (**d**). Basidiospores. (**e**). Stipitipellis elements. Scale bars: 10 μm.

**Figure 24 jof-09-00898-f024:**
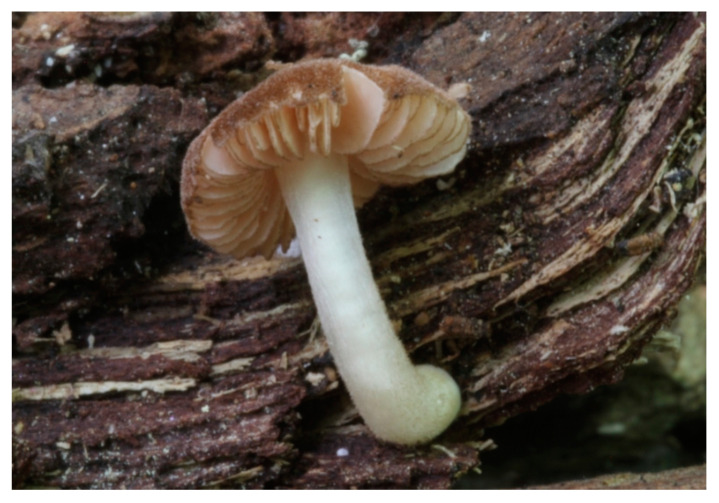
*Pluteus gausapatus* basidioma in situ, holotype BRNM 817745.

**Figure 25 jof-09-00898-f025:**
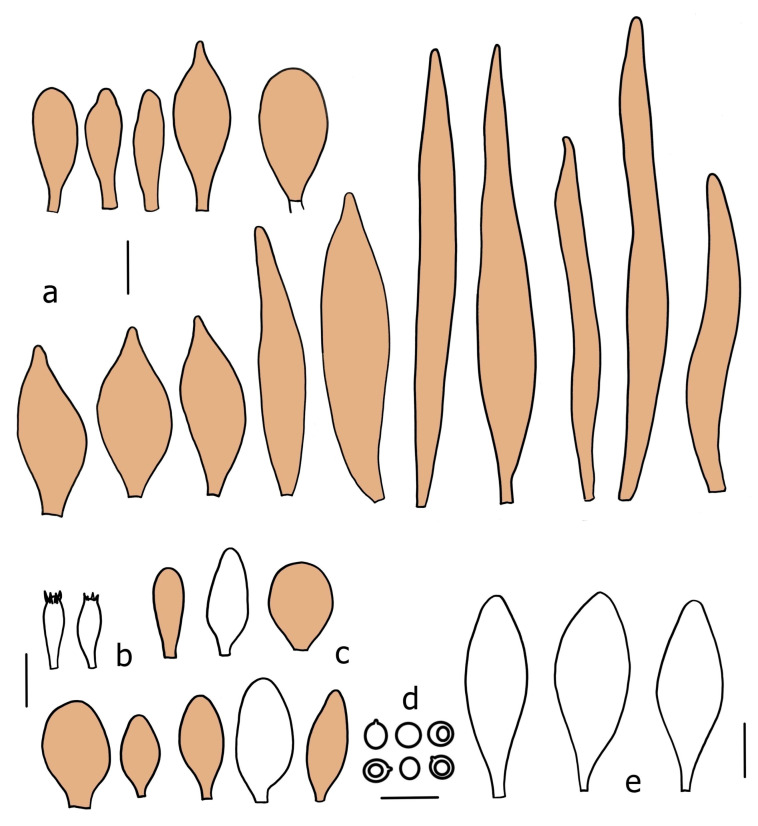
*Pluteus gausapatus*, holotype BRNM 817745. (**a**) pileipellis elements, (**b**) basidia, (**c**) cheilocystidia, (**d**) basidiospores, (**e**) pleurocystidia. Scale bars (**a**–**c**,**e**) = 20 μm, (**d**) = 10 μm.

**Figure 26 jof-09-00898-f026:**
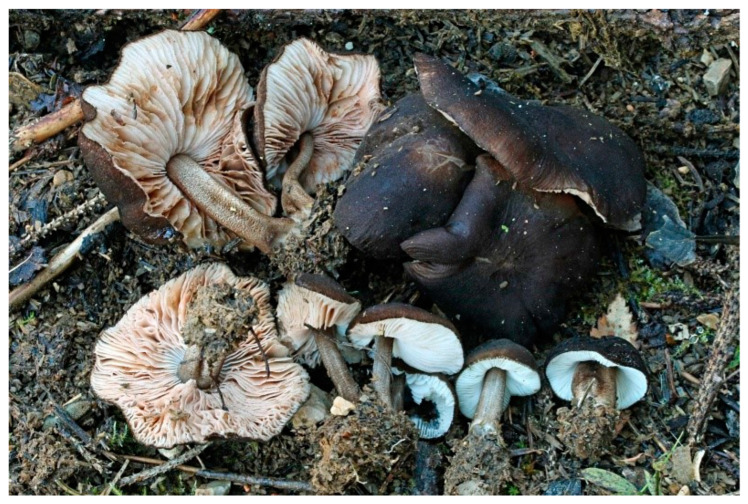
*Pluteus notabilis*, holotype PC0142588, basidiomata in situ.

**Figure 27 jof-09-00898-f027:**
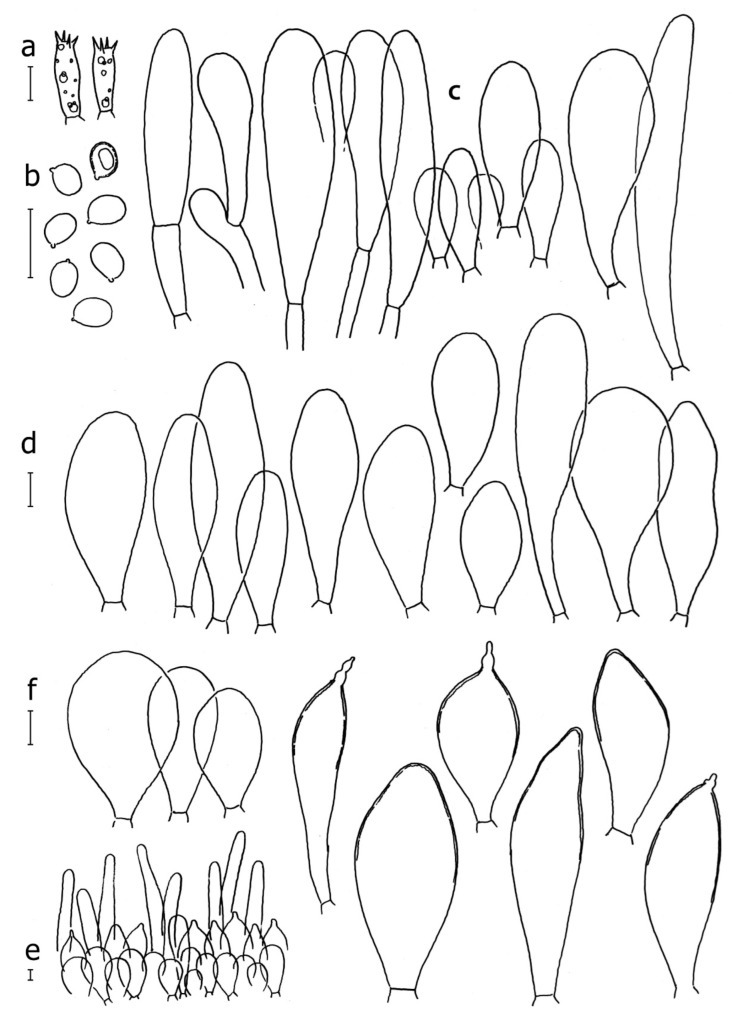
*Pluteus notabilis*, holotype PC0142588. (**a**) basidia. (**b**) basidiospores. (**c**) cheilocystidia. (**d**) pleurocystidia. (**e**) pileipellis. (**f**) pileipellis elements. Scale bars 10 µm.

**Figure 28 jof-09-00898-f028:**
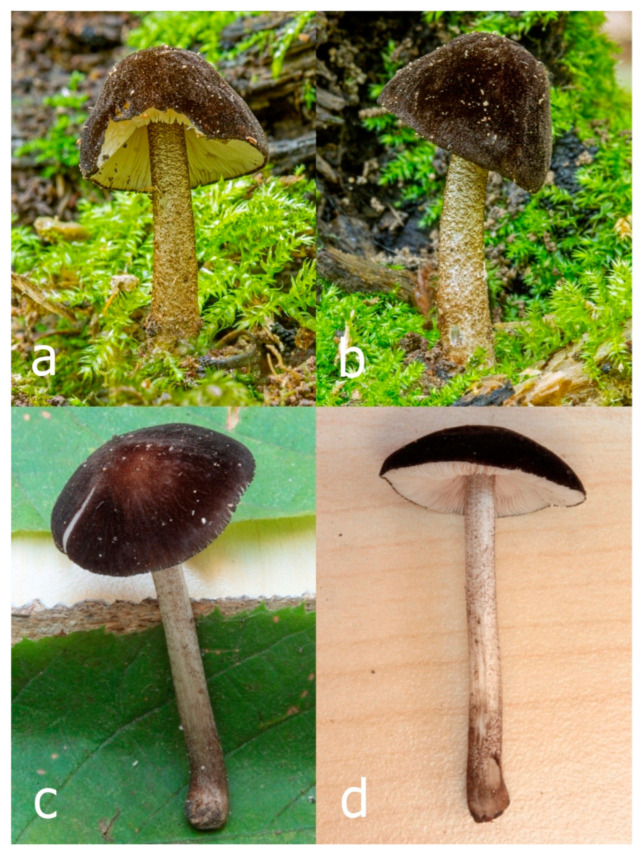
*Pluteus notabilis* var. *insignis*, basidiomata in situ (**a**,**b**) BRNM825868, (**c**,**d**) AG 22210280.

**Figure 29 jof-09-00898-f029:**
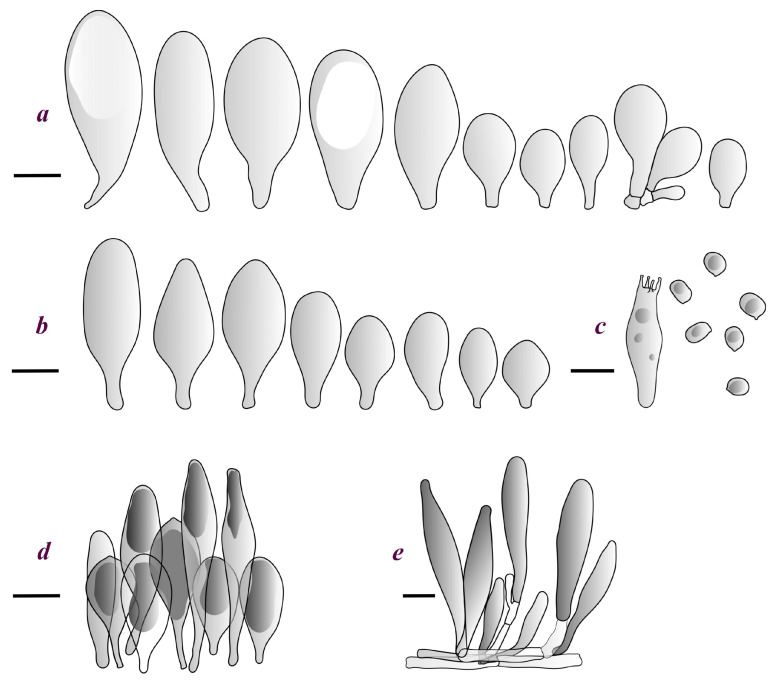
*Pluteus notabilis* var. *insignis* AG 22210280. (**a**) cheilocystidia, (**b**) pleurocystidia, (**c**) basidiospores and basidium, (**d**) pileipellis, (**e**) caulocystidia. Scale bars: 10 µm.

**Table 1 jof-09-00898-t001:** Sequences Used in the Study. Sequence codes refer to GenBank accessions unless otherwise specified (UNITE, BOLD, or iNaturalist repositories). Short ITS reads (ITS1-only or ITS2-only) are identified as such.

Taxon	Collection	Country	ITS	TEF1
*P. absconditus*	iNaturalist 128466549	USA (IN)	OP743482	-
*P. absconditus*	iNaturalist 129614045	USA (IN)	OP743435	-
*P. absconditus*	iNaturalist 16154325	USA (IN)	ON006999	-
*P. absconditus*	iNaturalist 112240775 (NBM-F-009786)	USA (TN)	OR229047	OR242143
*P. absconditus*	MO 136488 (NBM-F-009787)	USA (TN)	KM983689	OR242144
*P.* cf. *inflatus* (as *“podospileus”*)	JAC9765	New Zealand	MN738618	-
*P. crinitus*	FK2064	Brazil	KM983691	-
*P. cutefractus*	BRNM816221	Czech Republic	OR229051	OR242161
*P. cutefractus*	PRM 154563	Czech Republic	OR229052	OR242163
*P. cutefractus*	DMS 10213877	Denmark	OR229004 (ITS2-only)	-
*P. cutefractus* (as *“P. minutissimus”*)	MNHN-PC-FUSION111827	France	OR229080 (ITS1-only)	-
*P. cutefractus*	FG 09092018	Slovenia	OR229054	-
*P. cutefractus*	FG 17072019	Slovenia	OR229056	-
*P. cutefractus*	FG 26092015	Slovenia	OR229053	OR242164
*P. cutefractus*	FG12794	Slovenia	OR229055	-
*P. cutefractus*	MCVE 30110 (Holotype)	Slovenia	MN264751	-
*P. cutefractus*	MCVE 30111	Slovenia	MN264752	-
*P. cutefractus*	MCVE 30143	Slovenia	OR229057	-
*P. cutefractus*	BRNM825872	Spain	OR229050	OR242162
*P. cutefractus*	GM 3458	Spain	OR229048	OR242165
*P. cutefractus*	GM 3784	Spain	OR229049	-
*P. cystidiosus*	NBM-F-07400	Canada (NB)	OR229064	-
*P. cystidiosus*	HRL2941 (NBM-F-009791)	Canada (QC)	OR229041	-
*P. cystidiosus*	HRL713 (NBM-F-009792)	Canada (QC)	OR229065	-
*P. cystidiosus*	TNS-F 12398	Japan	HM562122	-
*P. cystidiosus*	LE 312852	Russia (Far East)	OR229063	OR242175
*P. cystidiosus*	LE 313335	Russia (Far East)	OR229062	OR242174
*P. cystidiosus*	AJ 782 (NBM-F-009790)	USA (MA)	KM983687	OR242171
*P. cystidiosus* (as “*P. seticeps*”)	MCBS 097	USA (MN)	MIN019-09 (BOLD)	-
*P. cystidiosus*	AJ 617 (NBM-F-009788)	USA (NY)	KM983686	OR242173
*P. cystidiosus*	AJ 619 (NBM-F-009789)	USA (NY)	KM983688	OR242172
*P. fuscodiscus*	BRNM766589	Czech Republic	OR229021	OR242154
*P. fuscodiscus*	BRNM817683	Czech Republic	OR229022	OR242153
*P. fuscodiscus*	TUF118497	Estonia	UDB017936 (UNITE)	OR242157
*P. fuscodiscus*	BRNM825871	France	OR229023	OR242155
*P. fuscodiscus*	DMS10334331	Iran	OR229027	-
*P. fuscodiscus*	FG02082015007	Italy	OR229026	-
*P. fuscodiscus*	FG11082017	Italy	OR229024	-
*P. fuscodiscus*	FG22082015	Italy	OR229025	-
*P. fuscodiscus*	LE 313448	Russia (European Part)	OR229019	OR242150
*P. fuscodiscus*	LE 312862	Russia (Far East)	OR229020	OR242149
*P. fuscodiscus*	LE 313336	Russia (Far East)	OR229018	OR242152
*P. fuscodiscus*	OK_TR_691	Turkey	OR229076 (ITS1-only)	OR242156
*P. fuscodiscus*	OK_TR_715	Turkey	OR229075 (ITS1-only)	OR242151
*P. granulatus* var. *tenellus*	G00126179	Switzerland	OR229003 (ITS2-only)	-
*P. inconspicuus*	PDD 72485	New Zealand	MN738614	-
*P. inexpectatus*	HRL3639 (NBM-F-009793)	Canada (QC)	OR229006	OR242145
*P. inexpectatus*	HRL4010 (NBM-F-009794)	Canada (QC)	OR229005	OR242146
*P. inflatus*	HRL3689 (NBM-F-009795)	Canada (QC)	OR229030	OR242130
*P. inflatus*	BRNM817761	Czech Republic	OR229033	OR242136
*P. inflatus*	BRNM825836	Czech Republic	OR229035	OR242132
*P. inflatus*	BRNM825837	Czech Republic	OR229036	OR242133
*P. inflatus*	BRNM825838	Czech Republic	OR229034	OR242134
*P. inflatus*	BRNM825839	Czech Republic	OR229032	OR242135
*P. inflatus*	PRM 154566	Czech Republic	OR229079 (ITS1-only)	-
*P. inflatus*	DMS 10027661	Denmark	OR229042	OR242129
*P. inflatus*	DMS 10041511	Denmark	OR229001 (ITS2-only)	-
*P. inflatus*	DMS 10205646	Denmark	OR229000 (ITS2-only)	-
*P. inflatus*	DMS 10206994	Denmark	OR228999 (ITS2-only)	-
*P. inflatus*	DMS10334330	Iran	OR229040	-
*P. inflatus*	LE 235208	Russia (European part)	OR229044	OR242128
*P. inflatus*	LE 262712	Russia (European part)	OR229045	OR242126
*P. inflatus*	LE 9809	Russia (European part)	OR229043	OR242127
*P. inflatus*	FG16092018026	Slovenia	OR229046	OR242139
*P. inflatus*	FG250620179	Slovenia	OR229002 (ITS2-only)	-
*P. inflatus*	BRNM825873	Spain	OR229031	OR242131
*P. inflatus*	GM3991	Spain	OR229037	
*P. inflatus*	Lundell 2541	Sweden	HM562196	-
*P. inflatus*	OKA 260	Turkey	OR229038	OR242141
*P. inflatus*	OKA 261	Turkey	OR229039	OR242142
*P. inflatus*	GM 1524	UK (England)	KM983690	OR242140
*P. inflatus*	iNaturalist 134948166	USA (WA)	iNaturalist 134948166	-
*P. inflatus* var. *alneus*	PRM 154261	Czech Republic	OR229028	OR242137
*P. inflatus* var. *alneus*	PRM 154569	Czech Republic	OR229029	OR242138
*P. minutissimus* f. *major*	G00126659	France	OR229077 (ITS1-only)	-
*P. minutissimus* f. *major*	G00126660	France	OR229078 (ITS1-only)	-
*P. phlebophorus*	AJ 81 (NBM-F-009110)	Spain	HM562039	ON133554
*P. podospileus*	BRNM825840	Czech Republic	OR229061	OR242166
*P. podospileus*	TUF118918	Estonia	UDB023496 (UNITE)	-
*P. podospileus*	LE 303682	Russia (South Siberia)	KX216331	OR242169
*P. podospileus*	LE 303687	Russia (South Siberia)	KX216332	OR242168
*P. podospileus*	LE 313589	Russia (South Siberia)	OR229060	OR242167
*P. podospileus*	LE 313598	Russia (South Siberia)	-	OR242170
*P. podospileus*	LE 313611	Russia (South Siberia)	OR229059	
*P. podospileus*	LE 313615	Russia (South Siberia)	OR229058	
*P. podospileus*	COFC-F 2929 (AJ 204, NBM-F-009808)	Spain	HM562049	-
*P. podospilloides*	LE 313230	Vietnam	MT611236	
*P. rugosidiscus*	BRNM761706	Slovakia	MH010876	LT991752
*P. seticeps*	HRL715 (NBM-F-009796)	Canada (QC)	OR229010	OR242159
*P. seticeps*	HRL716 (NBM-F-009797)	Canada (QC)	OR229009	-
*P. seticeps*	Shaffer798	USA (IL)	HM562199	-
*P. seticeps*	iNaturalist 129765552	USA (IN)	OP643122	-
*P. seticeps*	iNaturalist 132726783	USA (IN)	OP642974	-
*P. seticeps*	iNaturalist 143879657	USA (IN)	MK560139	-
*P. seticeps*	iNaturalist 15114066	USA (IN)	OP541808	-
*P. seticeps*	iNaturalist 31152914	USA (IN)	ON006967	-
*P. seticeps*	iNaturalist 91406034	USA (IN)	OM972589	-
*P. seticeps*	iNaturalist 91544459	USA (IN)	OM809392	-
*P. seticeps*	iNaturalist 91684267	USA (IN)	OM809298	-
*P. seticeps*	iNaturalist 91684550	USA (IN)	OM809293	-
*P. seticeps*	iNaturalist 91684832	USA (IN)	OM809282	-
*P. seticeps*	iNaturalist 91977585	USA (IN)	OM972279	-
*P. seticeps*	iNaturalist 92040615	USA (IN)	OM972280	-
*P. seticeps*	iNaturalist 92139428	USA (IN)	OM972303	-
*P. seticeps*	SF23	USA (MO)	HM562191	-
*P. seticeps*	iNaturalist 88771421 (NBM-F-009801)	USA (NJ)	OR229014	-
*P. seticeps*	iNaturalist 91066075 (NBM-F-009802)	USA (NJ)	OR229015	-
*P. seticeps*	iNaturalist 93920427	USA (NY)	OM441921	-
*P. seticeps*	MO 177404 (NBM-F-009809)	USA (OH)	OR229017	OR242160
*P. seticeps*	MO 324271 (NBM-F-009810)	USA (PA)	OR229016	OR242158
*P. seticeps*	iNaturalist 112235090 (NBM-F-009798)	USA (TN)	OR229013	-
*P. seticeps*	iNaturalist 112237607 (NBM-F-009799)	USA (TN)	OR229012	-
*P. seticeps*	iNaturalist 112335052 (NBM-F-009800)	USA (TN)	OR229011	-
*P. seticeps*	SF24	USA (WI)	HM562192	-
*P. seticeps* var. *cystidiosus*	R.L. Homola 1340 (MICH 69572)	USA (MI)	OR229066	-
*Pluteus brunneocrinitus*	MC4535	Brazil	KM983692	
*Pluteus gausapatus*	BRNM817745	South Korea	OR229067	OR242177
*Pluteus millsii*	AJ838 (NBM-F-009803)	USA (CT)	KM983694	-
*Pluteus millsii*	iNaturalist 34927657	USA (KY)	OK586458	-
*Pluteus millsii*	iNaturalist 112187791 (NBM-F-009804)	USA (TN)	OR229068	OR242182
*Pluteus millsii*	MO 458003 (NBM-F-009805)	USA (TN)	OR229069	OR242183
*Pluteus necopinatus*	FK 1701	Brazil	KM983693	
*Pluteus notabilis* var. *insignis*	AG 22210280	Italy	OR229073	-
*Pluteus notabilis* var. *insignis*	BRNM825868	Slovakia	OR229072	OR242180
*Pluteus notabilis* var. *insignis*	BRNM825867	Spain	OR229081 (ITS1-only)	OR242181
*Pluteus notabilis* var. *notabilis*	BRNM825870	Czech Republic	OR229070	OR242178
*Pluteus notabilis* var. *notabilis*	PC0142588	France	OR229071	OR242179
*Pluteus* sp.	GDGM41576	China	KU382738	-
*Pluteus* sp.	GM 3751	France	OR229007	new
*Pluteus* sp.	Soil Sample	Mexico	UDB05068676 (UNITE)	-
*Pluteus* sp.	Soil Sample	Mexico	UDB05068682 (UNITE)	-
*Pluteus* sp.	Soil Sample	Mexico	UDB05068679 (UNITE)	-
*Pluteus* sp.	JAC 15123	New Zealand	MN738664	-
*Pluteus* sp.	iNaturalist 27406926 (NBM-F-009806)	USA (IN)	ON006984	OR242176
*Pluteus* sp.	iNaturalist 13936381	USA (MS)	OP541701	
*Pluteus* sp.	MO 270622 (NBM-F-009807)	USA (OH)	OR229008	OR242147

## Data Availability

Publicly available datasets were analyzed in this study. The data can be found here: https://www.ncbi.nlm.nih.gov, accessed 30 August 2023; https://www.mycobank.org, accessed on 30 August 2023.

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
