# Peer review of "Holarctic Species in the Pluteus podospileus Clade: Description of Six New Species and Reassessment of Old Names"

_jof, 2023, doi:10.3390/jof9090898_

Round 1

Reviewer 1 Report

Notes on ms on Pluteus podospileus group

Thank you for another thorough paper on Pluteus species.

In general this looks like a good paper, but as the descriptions and illustrations are made by different authors, uniformity in some aspects is lacking. Please compare the various parts with each other and straighten things out.

There are also many data still missing - for instance in the colour figures, Genbank and Mycobank numbers, and herbarium data. 

I would like to see information on the value of the ITS sequences - a tree would be very welcome, and also whether this part of the genome is enough to distinguish between species. Does ITS1 alone or ITS2 alone work as well? As nanopore sequence data are used this is important information to add. 

My notes are in order of the paper, and some are just spelling, others in depth remarks.

Author information: please make sure that there is uniformity among the entries - for instance a city name is lacking for Malysheva’s address, a country is absent for Ida Broman Nielsen, Italian and English are mixed in the names of the institutes of Dovana, etc.

Morphology - several different colour designations are used, not only the RAL Design colors. so please put that under Material and Methods [this is one example of the non-uniformity among the parts].

Line 142 - ;please reword (now it reads that Collections were performed…)

Could you indicate in the table 1 with sequence data which ITS sequences were only the short read (by nanopore NGS)? This is important to know!

Line 167 - The tweezer was thoroughly cleaned between samples [not ‘between each sample’

Line 197 - the criteria used for selecting the sequence have to be given in much more detail than here. and please don’t use the word ‘picking’ 

Table 1 - Genbank numbers are missing.

and it is not acceptable to reference to an iNaturalist entry for the sequence data.

what is MIN019-09 (BOLD)?

why is (Unite) added to UDB023496?

Line 240 - add ‘and’ before the TEF1-alpha 

Question about sampling efforts - there is only one collection from western North America (P. inflatus from WA) included in this study. Does this mean that this group does not occur there, or that your sampling effort was not sufficient ? I would like to see this discussed in the paper.

Line 272 - first word ‘or’ [not ‘of’]

All the colour figures lack a b c, etc.

The information in the first parts of the species (the nomenclature part) is not uniform, please put them side by side and compare! 

Please add for misapplications of the name by whom these names were misapplied. that is useful information to have ! (example : Pluteus podospileus under P. seticeps), and make a distinction between misapplied name and excluded interpretations.

Lines 437–444 - Is Pluteus inflatus var. alneus a synonym? if so, put it in the list of synonyms.

Start the list of synonyms with Pluteus granulatus var. tenellus, as it was published before P, minutissimus f. major and is at higher rank. These lists of synonyms should list the available names in sequence of priority.

Line 655-656 - please give full reference to the publication of Pluteus cutefractus - compare with other species.

Line 818 - Pluteus psichiophorus var. seticeps is a synonym of Pluteus seticeps, not a misapplied name. 

If Singer used the name for another taxon, his interpretation has to be excluded  (put ‘Excluded before the reference, and indicate which species his interpretation is), but he did make the combination, whatever his interpretation was.

line 449 Kühner [with an ‘umlaut’ on the u]

spelling concolourous - compare with color and coloured - please be consistent, either British English, or American-English, but not both.

and check words like center / centre - both are used here.

Line 495 - under Distribution also say that this species was found in western North America.

Line 581 - the collection number does not correspond with that in the lists and tables.

Line 716 - ‘Pluteus cutefractus was described as having’ ..’[not ‘by having’]

Line 724 - fuscodiscus refers to the dark central disc on the pileus, not to a darker dic. please always give the meaning of the epithet first, then why you chose the name.

Lines 857-868 - add herbarium numbers, to replace the ***

Line 950-966 - why are some collection data in bold?

Figure 5 -the spores are minuscule in this figure

Figure 19 - the spores are minuscule in this figure - much too small….

Figure 21 - the spore figure is much too small.

line 1070 and in other places- remove the space between S. and D. Russell.

Pluteus millsii - the photos do not show a woolly stipe. These descriptive words of the stipe surface are all very subtle, and have to be compared with each other in this paper.

Line 1082 - ah, were these specimens collected during the MSA meeting in New Haven? 

Pluteus notabilis - line 1221 - which part of the fruitbodies have these flocculose edges? this hangs a bit in a vacuum.

line 1224 - reread the habitat part where the holotype was found and reword.

line 1286 - finally (not finelly)

line 1299 - add µm after the avl x avw sizes

lines 702 & 1323 - ‘Mediterranean ambient’: what do you mean?

Lines 1362–1363  - here i see a comparison, but not with which species :P. praestabilis is compared. Please be explicit.

Line 1391-1392 – and other parts of this discussion - i miss more concrete information which characters do define those species. Here it is said that a combination has to be used, but it is not specified which characters. In this group where all species look very much alike, more specificity is needed, if you really want to distinguish all those taxa.

Line 1396 - ‘be’ instead of ‘by

Line 1400 - is the average Q of the spores 1.03-1.07? if so, please add the * to Q

Line 1432 - used [not use]

Line 1443-1444 - Pluteus aff. podospilloides [no capital and 2 l’s]

Line 1448 - understanding [not understading]

Line 1444 - only eastern North American species have been discovered - there is one collection from Washington. In general, the mycoflora of eastern North America differs greatly from that in western North America, due to climate, geological history, and vegetation.

lamella edge or lamellar edge  - please choose one and stick to it (compare the usage in the two keys!)

Key - please do not use an entry that is vague, such as ‘Some features different’ 

line 1494 - mucronate elements, mucro up to 8 µm long [be explicit here, as written the size refers to the whole element, not just the tip]

see my notes above

Author Response

Please see the responses in the attachment

Reviewer 2 Report

I reviewed the manuscript entitled Holarctic Species in the Pluteus podospileus Clade. Six New Species Described and Old Names Reassessed. Overall, the manuscript is well written and contains a lot of important data and useful information for readers. In my opinion, the manuscript could be published in Journal of Fungi.

The ABSTRACT is well written (the aim, methods and results of the research are briefly presented).

The INTRODUCTION is well written and slowly introduces readers to the research topic.

The Materials and Methods, RESULTS AND DISCUSSION are very well presented.

In the references section, check all the sources, as I notice that they are not harmonized. For example, somewhere the full name of the journal is written, and somewhere it is an abbreviation, sometimes it is in italics, and sometimes it is not. Prepare a list of references in accordance with the journal's instructions.

I suggest this manuscript to be published in Journal of Fungi after the recommended minor technical corrections.

Author Response

(The authors gave the same response as above.)

Reviewer 3 Report

This manuscript "Holarctic Species in the Pluteus podospileus Clade. Six New Species Described and Old Names Reassessed" is a fairly well written paper worth to be considered in JOF but it needs some revisions. The title needs revising. Authors are advised to check the annotated comments in the PDF attached and revise the manuscript properly.

Overall, the English is fine. 

Author Response

(The authors gave the same response as above.)
